 eLIFE

# Reconstitution of self-organizing protein gradients as spatial cues in cell-free systems

**Katja Zieske\*, Petra Schwille\***

Department of Cellular and Molecular Biophysics, Max Planck Institute of Biochemistry, Martinsried, Munich, Germany

**Abstract** Intracellular protein gradients are significant determinants of spatial organization. However, little is known about how protein patterns are established, and how their positional information directs downstream processes. We have accomplished the reconstitution of a protein concentration gradient that directs the assembly of the cell division machinery in *E.coli* from the bottom-up. Reconstituting self-organized oscillations of MinCDE proteins in membrane-clad soft-polymer compartments, we demonstrate that distinct time-averaged protein concentration gradients are established. Our minimal system allows to study complex organizational principles, such as spatial control of division site placement by intracellular protein gradients, under simplified conditions. In particular, we demonstrate that FtsZ, which marks the cell division site in many bacteria, can be targeted to the middle of a cell-like compartment. Moreover, we show that compartment geometry plays a major role in Min gradient establishment, and provide evidence for a geometry-mediated mechanism to partition Min proteins during bacterial development.

\*For correspondence: zieske@biochem.mpg.de (KZ); schwille@biochem.mpg.de (PS)

**Competing interests:** The authors declare that no competing interests exist.

**Reviewing editor**: Mohan Balasubramanian, University of Warwick, United Kingdom

## Introduction

Intracellular concentration gradients of proteins in micrometer-sized cells have long been thought to be unsustainable due to diffusion. However, the past decade revealed the existence of multiple intracellular gradients in eukaryotes and prokaryotes and their significance in providing positional information within the cellular compartment (*Kiekebusch and Thanbichler, 2014*). Although these protein gradients are now emerging as significant general motifs to spatially organize cells, little is known about how protein patterns are established by local unmixing, and maintained on a molecular level. What are the ultimate cues and mechanisms to localize gradient forming proteins? Are protein gradients modulated by boundary conditions of the cell and how are downstream proteins directed by the positional information of protein gradients?

Cellular reconstitution methods enable us to study biological processes under defined conditions, and have significantly contributed to our understanding of molecular interactions and kinetics of proteins in simplified environments. To systematically investigate the mutual dependence between biochemical networks and three-dimensional cellular organization several techniques to confine proteins in micro compartments have recently been devised. One approach to achieve three-dimensional confinement of proteins and cytoplasmic extracts is by their encapsulation in water-oil droplets or vesicles. (*Pinot et al., 2009, 2012*; *Good et al., 2013*; *Hazel et al., 2013*) These systems add tremendously to our understanding of how a constrained reaction space, and varying surface-to-volume ratios affect (bio) chemical reactions. Applying subtle mechanical forces to these systems, they may be also deformed into non-spherical shapes. However, if more complex geometries need to be realized, or if the compartments need to be mechanically stabilized, biochemical systems may also be reconstituted on and in spatially tailored microfabricated environments (*Garner et al., 2007*; *Laan et al., 2012*). In particular, combinations of two- and three-dimensionally engineered substrates with lipid membranes

**eLife digest** When a cell divides, it is important that its contents are separated in the right place to make sure that both daughter cells have everything that they need to survive. To do this, the molecular 'machinery' that physically divides the cell needs to know where to assemble.

In the bacterium *E. coli*, the location of cell division depends on a group of proteins called the Min proteins. These proteins are not evenly distributed over the cell. Instead, they oscillate back and forth to set up concentration gradients, with the concentration of Min proteins being lowest in the middle of the cell and highest at the ends. The machinery that divides the cell assembles at the point where the concentration of Min proteins is lowest. However, it is not clear exactly how the protein gradients are set up in the cell, and whether these gradients are indeed sufficient to position the cell division machinery.

To explore this process, Zieske et al. engineered artificial cells that mimicked some of the basic properties of living cells. In these artificial cells, the Min proteins organized themselves into gradients that were similar to those found in living cells. This gradient then caused another protein called FtsZ—which is involved in cell division—to accumulate in the middle of the artificial cell. Zieske et al. also showed that the shape of the artificial cell influenced the shape of the protein gradient.

This research shows that the interplay between the shape of a bacterial cell and a defined set of proteins could control the position of cell division. The simplified system that Zieske et al. have developed could also be used to study other aspects of cell organization and cell division.

have been applied to study the role of simple membrane geometries on membrane interacting protein networks in vitro (*Schweizer et al., 2012*; *Zieske and Schwille, 2013*). However, although such in vitro experiments are now being successfully applied to identify the minimal components and interactions of protein networks, we are still far from understanding the mutual interdependence of protein functionalities and physical parameters in generating nonhomogeneous protein distributions for providing positional information within the cellular compartment. Therefore, further development of bottom-up approaches are required to tackle the minimal requirements for nonhomogeneous protein localization. Studying gradient formation in a simplified and controlled environment, detached from the complexity of living cells, should shed light on the role of individual biochemical and physical parameters for regulated protein localization in space and time.

Among the most fundamental and significant gradients in cells are those that set the spatial cues for cytokinesis. In many bacteria, and also in eukaryotes that undergo symmetric division, protein gradients with the lowest concentration in the middle of the cell and the highest concentration at the cell poles have been identified. These gradients target the division site through polar inhibition of the division machinery assembly. For instance, in fission yeast, a Pom1 gradient regulates the onset of mitosis (*Celton-Morizur et al., 2006*; *Padte et al., 2006*; *Martin and Berthelot-Grosjean, 2009*; *Moseley et al., 2009*). Similarly, in many bacteria, the assembly of the conserved ring-forming cell division protein FtsZ is inhibited at the cell poles by protein gradients of FtsZ inhibitors. In *Caulobacter crescentus*, a gradient of the cytokinesis inhibiting protein MipZ is established, (*Thanbichler and Shapiro, 2006*) whereas in *Bacillus subtilis*, an inhibitory complex of MinC and MinD is attached to the cell poles by MinJ and DivIVA. (*Edwards and Errington, 1997*; *Bramkamp et al., 2008*; *Patrick and Kearns, 2008*) Comparably, *Escherichia coli* employs the membrane-targeted MinC/MinD complex to inhibit FtsZ assembly at the cell poles. (*Lutkenhaus, 2007*) Interestingly however, MinC and MinD in *E.coli* do not form a static gradient, but the Min protein patterns oscillate from pole to pole. (*Raskin and de Boer, 1999b*; *Raskin and de Boer, 1999a*) These dynamic oscillations require ATP as an energy source for the ATPase MinD and MinE, which accelerates hydrolysis of ATP by MinD. (*de Boer et al., 1991*; *Hu and Lutkenhaus, 2001*) On time-average, the oscillations result in an effective concentration gradient of the MinC/MinD complex.

While MinD and MinE have been identified as a minimal set of proteins that is able to self-organize into dynamic pattern on flat supported membranes, (*Loose et al., 2008*) the minimal requirements for gradient formation, and the mechanisms of forwarding positional information to downstream processes are still controversial and only begun to be elucidated. Labeling MinE and co-reconstituting MinE with MinD in cell-shaped compartments, we demonstrated that pole-to-pole oscillations of the

Min system can be reconstituted in vitro (*Zieske and Schwille, 2013*). However, time-averaged Min protein gradients in vitro have not been reported to date and thus, direct evidence for the establishment of steady protein gradients by self-organizing oscillations of MinD and MinE without additional proteins is still missing.

Moreover, it is still ambiguous how cell geometry affects gradient formation. On the one hand, live cell experiments with cell shape mutants, as well as in vitro reconstitutions, have shown that compartment geometry affects pattern formation of the Min proteins (*Corbin et al., 2002*; *Varma et al., 2008*; *Raskin and de Boer, 1999b*; *Zieske and Schwille, 2013*). On the other hand, protein gradients should be robust against morphological changes during cell development and growth. Living bacteria typically double their length during one live cycle and gradually constrict during cell division. Thus, they undergo significant changes in cell geometry, which should not disturb the establishment and functional role of Min protein gradients. Whether cell shape affects pattern formation only if the geometry of a cell is artificially perturbed, or whether a cell might use its own geometry as a readout and control parameter to shape protein gradients, still has to be determined. Thus, the identification of principal cues for gradient establishment, and their modification by spatial parameters, such as the three-dimensional geometry of a cell, remains an outstanding challenge.

In this study, we were able to establish, from a system of soluble and initially well-mixed proteins, an effective steady protein concentration gradient, potent of targeting the assembly of cell division proteins to the middle of artificial membrane-clad compartments. In particular, FtsZ, which marks the cell division site and represents the cytoskeletal backbone for the divisome machinery in many bacteria, can be targeted to the middle of a cell-like compartment by a gradient of Min proteins. Moreover, we demonstrate that orientation and partitioning of Min protein gradients is controlled by compartment geometry. Notably, our synthetic system opens a new way to study complex organization principles in a simplified environment and provides novel insights into the basic biophysical and biochemical requirements for gradient formation.

## Results

### Reconstitution of protein gradients in cell-shaped compartments

To determine the minimal requirements for gradient formation, we investigated the spatially self-organizing Min system as a prototype for gradient formation in a synthetic environment (*Figure 1A*). We first enclosed purified MinD and MinE proteins from the bacterium *E. coli* and ATP in membrane clad soft polymer compartments of picoliter sample volume and cell-shaped geometry (*Figure 1—figure supplement 1*, *Figure 1—figure supplement 2*). Consistent with previous results and live-cell studies, MinE and MinD dynamically oscillate from pole-to-pole. Previous controls with purified Min proteins on flat bilayers and the non-hydrolysable ATP analog ATPγS confirmed that the dynamic protein patterns were thereby powered by ATP (*Loose et al., 2008*).

Compared to living cells, the temporal scale for the oscillation period is conserved in the synthetic system. However, both the compartments and the spatial scales of the protein waves are about ten times bigger as compared to living cells. Why the Min patterns in vivo are about ten times larger than in vitro is quantitatively not fully understood. Different protein ratios, ionic strength of the buffer and membrane composition have been shown to affect the spatial scales (*Loose et al., 2008*; *Vecchiarelli et al., 2014*). Furthermore the membrane potential (*Strahl and Hamoen, 2010*) and molecular crowding in a living cell could influence Min protein patterns. An advantage of the larger spatial scales in vitro is the possibility to study with much greater detail how the concentration profiles of MinD and MinE, which result in the formation of MinD gradients, are modulated during an oscillation period. Following the MinD and MinE concentration profiles along the compartment length over time, we found that the dynamic profiles are characterized by the following succession of events: First, attachment and increase in concentration of MinD at one site of the compartment, then delayed attachment of MinE to MinD and increase in MinE intensity at the trail of MinD (*Figure 1—figure supplement 3*). Although the size of bacterial cells and photo bleaching is limiting for characterizing the Min profiles in vivo in such detail as it is possible in vitro, the intensity peak of MinE at the trail of the Min pattern was also observed in *E. coli* and is generally referred to as 'MinE ring'. (*Raskin and de Boer, 1997*; *Fu et al., 2001*) After formation of the MinE peak MinD/E were disassembled from the membrane at the trail of the MinD zone, which was accompanied by assembly of a new MinD/E zone at the other pole of the compartment (*Figure 1—figure supplement 3*). This disassembly of Min protein patterns

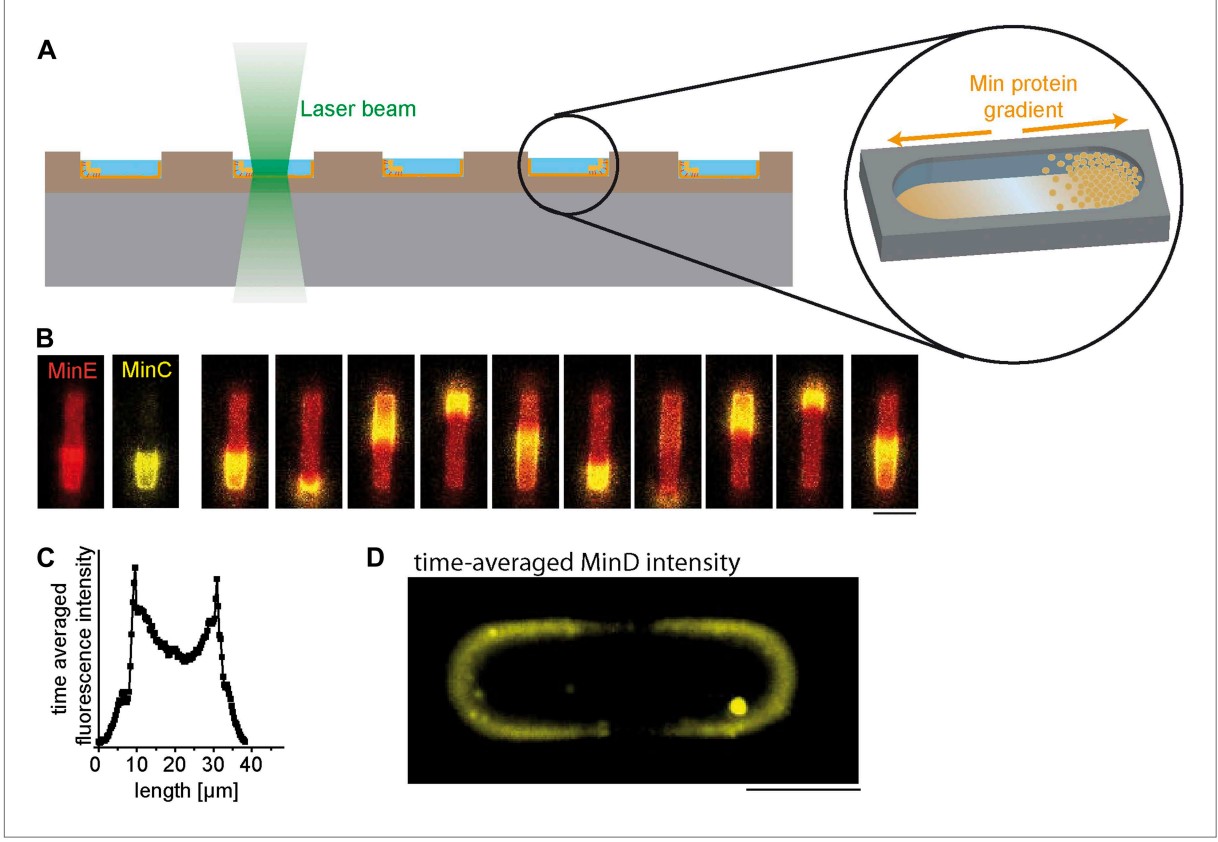

**Figure 1**. Protein gradients self-organize in soft-polymer containers. (**A**) Experimental setup. Purified proteins of the MinC/D/E system were reconstituted in membrane-clad soft-polymer compartments and imaged by confocal microscopy. Profiles and electron micrographs of the compartments are shown in *Figure 1—figure supplement 1* and a more detailed description of the assay to reconstitute Min protein oscillations is presented in *Figure 1—figure supplement 2*. *Figure 1—figure supplement 3* demonstrates how the intensity profiles of MinD and MinE along the length axis of the compartments are modulated with time. To mimic a bacterial membrane we used *E. coli* polar lipids to generate supported lipid membranes. A more detailed description of how lipid composition influences pattern formation of Min proteins is presented in *Figure 1—figure supplement 4*. (**B**) In cell-shaped compartments, eGFP-MinC (yellow) follows the oscillations of MinD and MinE (red). Comparable results were obtained in more than a hundred soft-polymer compartments. Confocal time-lapse images with the image plane at the bottom of the compartment. Protein concentrations: 1 µM MinD, 0.9 µM MinE, 0.1 µM MinE.Atto655 (red), 0.05 µM eGFP-MinC (yellow). Time between individual frames: 30 s. Scale bar: 5 µm. (**C**) The time-averaged concentration profile of MinC along the long axis of a compartment has a distinct concentration minimum in the middle of the compartment. The time averaged distribution of protein concentrations was calculated by acquiring time-lapse-images with the focal plane at the middle of the compartment and averaging the intensity of the acquired frames. (**D**) Time-averaged fluorescent signal of eGFP-MinD with image plane in the middle of a compartment. An intensity offset is subtracted to better visualize the gradient along the boundary of the compartment. 0.9 µM MinD, 0.1 µM eGFP-MinD, 1 µM MinE. Scale bar: 5 µm. Stable pole-to-pole oscillations which result in the time-averaged gradient of MinD are severely affected if the membrane targeting sequence of MinE is deleted (*Figure 1—figure supplement 5*).

The following figure supplements are available for figure 1:

**Figure supplement 1**. Micro compartments in a soft-polymer chip.

**Figure supplement 2**. Experimental setup.

**Figure supplement 3**. Protein gradients in vitro are established by dynamic redistribution of proteins.

**Figure supplement 4**. Lipid composition dependent formation of protein gradients.

**Figure supplement 5**. Deletion of the membrane targeting sequence of MinE affects pattern formation in micro compartments.

at the trailing zone during each oscillation is comparable to the disassembly at the trailing edge of travelling Min patterns on flat membranes. The main difference of Min proteins profiles on flat membranes and in microcompartments is that the concentration profiles on flat membranes only change their localization while traveling across the membrane. In contrast, during oscillations in compartments new membrane attached Min protein patterns need to be repeatedly established and disassembled every half oscillation period, which results in a remarkable remodelling of the concentration profiles during the oscillation cycle and an asymmetry in their rise and decay phases at the poles.

To determine the mean spatial distribution of MinD, we averaged the fluorescence signal of MinD within the cell-shaped compartments over time using confocal time-lapse movies. Remarkably, MinD formed indeed a clear nonhomogeneous concentration profile, with the lowest concentration in the middle of the compartment (*Figure 1D*). Thus, we conclude that using the Min system, the establishment of an effective protein gradient indeed requires only a lipid membrane, an appropriate reaction space, and two proteins that self-organize under the consumption of energy, in the form of ATP.

While the requirements for Min gradient formation are now understood to a point, at which these gradients can be reconstituted in vitro, less is known about the importance of the structural properties of the gradient forming proteins. Although MinD has been shown to interact with membranes in vitro and MinE interacts with MinD (*Hu et al., 2002*), MinE also harbours a short membrane targeting sequence at its N-terminus (*Park et al., 2011*). Mutations in the membrane targeting domain of MinE result in defects in cell division and in distinct phenotypes, such as filamentous cells or a minicelling phenotype, respectively (*Park et al., 2011*). However how the membrane targeting of MinE contributes to a normally functioning Min system is not fully understood. To gain a deeper insight in how the membrane targeting sequence of MinE contributes to gradient formation of the Min system we purified and labelled a MinE mutant without its membrane targeting sequence (MinE without amino acids three to eight, herein referred to as MinE(Δ3–8)) and reconstituted it with MinD in microcompartments. Interestingly, MinE(Δ3–8) resulted in a severe phenotype in the bottom-up system. The Min system was still able to self-organize into dynamic pattern, but instead of a stable pole-to-pole oscillation, the Min system formed dynamic patches with irregular shape and without a constant axis of movement (*Figure 1—figure supplement 5*). Thus, although MinE(Δ3–8) still counteracted MinD, no stable MinD gradients were generated. Depending on the degree of affecting the membrane targeting sequence of MinE, the effective generation of a gradient and its localization along the long axis of the cell might therefore be altered to different degrees, which might account for the distinct phenotypes in vivo, such as filamentous cells or mini cells.

While the structural properties and the diffusion-reaction driven self-organization of MinD and MinE determine the non-equilibrium distribution of the Min system, a third protein, MinC, is proposed to be the actual mediator to forward positional information of the Min gradient to downstream targets. In particular, MinC is proposed to inhibit FtsZ–the cytoskeletal framework of divisome assembly (*Hu et al., 1999*; *Hu and Lutkenhaus, 2000*; *Shen and Lutkenhaus, 2009, 2010*). To investigate the spatial distribution of eGFP-MinC in our minimal system, we co-reconstituted MinD and MinE oscillations with MinC in micro compartments. We found that eGFP-MinC oscillated from pole-to-pole by coupling to MinD/E patterns, which strongly resembles the dynamics of MinC in vivo (*Figure 1B*). Based on time-lapse movies, we then analyzed the time-averaged distribution of eGFP-MinC and found the concentration profile of MinC also exhibit a pronounced minimum in the middle of the compartment (*Figure 1C*). This experiment revealed that the effective non-equilibrium distribution of the complete Min system, with the highest concentration of MinC at the cell poles, is fully recapitulated,in our simple cell-free setting.

Note that the *E. coli* lipid extract for generating membranes in these experiments is a mixture of mainly three different lipids, (phosphatidylethanolamine [PE, neutral], phosphatidylglycerol [PG, negatively charged] and cardiolipin [CA, negatively charged]) and that the membranes have a net negative charge. Recently it has been shown that a minimal lipid bilayer for pattern Min formation on flat supports only requires two different lipids. Moreover, cardiolipin, which has previously been proposed as a structural polar cue, which might help in triggering the pole to-pole oscillations of Min proteins, was not required for Min pattern formation on planar supports. Instead, the negative charge has been suggested to play a primary role in formation of surface waves on planar membranes. (*Vecchiarelli et al., 2014*) To address whether membrane charge and not the chemical membrane composition is also sufficient for gradient forming pole-to-pole oscillations, we established the minimal membrane composition for Min protein oscillations in our synthetic system. First, we confirmed the aforementioned

results that membrane charge is a strong determinant for Min pattern formation on flat supported membranes (*Figure 1—figure supplement 4A–D*). Then we reconstituted the Min proteins in compartments with two different minimal membrane mixtures: First, with 70% neutrally charged DOPC and 30% negatively charged cardiolipin, and then with 70% DOPC and 30% negatively charged PG. In both cases, the proteins retained their ability of pattern formation and pole-to-pole oscillations (*Figure 1—figure supplement 4E*). Thus, we conclude that Min gradients can be established without any structural cues in the membrane. In contrast, simple physical parameters, such as electrostatic interactions between the membrane and the proteins, are strong determinants for gradient formation. For high amounts of negatively charged lipids, the length scales of the protein patterns decrease (*Figure 1—figure supplement 4B* and [*Vecchiarelli et al., 2014*]). Thus, the spatial scales of the Min proteins on highly negatively charged membranes would be too small to form stable pole-to-pole oscillations in compartments, which were engineered for protein length scales on *E.coli* membranes. The implication that Min protein oscillations are perturbed if the membrane charge increases is in agreement with live cell studies of GFP-MinD in PE-lacking *E.coli* cells, which demonstrate that Min protein pattern are affected if the membrane charge is increased (*Mileykovskaya et al., 2003*) and provides further evidence that a balanced ratio of charged and non-charged lipids is of critical importance for the live cycle of *E.coli*.

## Localization of downstream targets by reconstituted protein gradients

Intracellular protein gradients play a pivotal role in cellular organization through providing positional information for downstream proteins. However, although protein gradients are now emerging as general motives to organize cells, tools to study gradient-mediated organization of three-dimensional space are limited. To directly investigate whether the established Min gradient indeed provides an efficient cue to position downstream targets, we attempted the assembly and positioning of FtsZ to the middle of a synthetic compartment in a co-reconstitution assay. To keep the system as minimal as possible, we avoided the native membrane adaptors of FtsZ, FtsA and ZipA, and used a FtsZ hybrid protein, fused to YFP and a membrane targeting sequence (FtsZ-mts) (*Osawa et al., 2008*; *Arumugam et al., 2014*). In spite of this simplification, we found that highly FtsZ-mts enriched regions were clearly visible in the center of the compartments, while FtsZ-mts localization was strongly reduced at the polar regions (*Figure 2*). Negative controls with FtsZ-mts bundles on flat supported bilayers confirmed that all thee Min proteins are required to deplete FtsZ-mts from the membrane (*Figure 2—figure supplement 1*) (*Arumugam et al., 2014*). This unequivocally demonstrates that an effective time-averaged protein gradient can indeed regulate the localization of down-stream targets to predefined localizations, such as the middle of a compartment.

In contrast to the living cell, the FtsZ-mts distribution in the middle of synthetic compartments appears to be rather broad. The major reason for the wide FtsZ-mts distribution is most likely the ten times larger Min protein pattern in vitro, as compared to living cells. The wide MinC minimum allows multiple FtsZ-mts bundles to localize in a wide region in the middle of the compartment.

Note that FtsZ-mts bundles in the central region aligned perpendicular to the long axis of the compartments (*Figure 2A,B,D*). This finding is in agreement with previous results that demonstrated a preferential alignment of FtsZ-mts along negatively curved membranes on grooved glass as membrane support. (*Arumugam et al., 2012*).

## The MinC-G10D mutant interacts with the MinD/E system, but does not inhibit FtsZ-mts at compartment poles

Positional information of the MinD gradient in vivo and in our synthetic system is mediated by the FtsZ inhibitor MinC which directly interacts with FtsZ (*Hu et al., 1999*). MinC has a C-terminal and an N-terminal domain, (*Hu and Lutkenhaus, 2000*; *Cordell et al., 2001*) which are both involved in FtsZ inhibition. The C-terminal domain of MinC is responsible for binding to MinD and dimerization of MinC. Furthermore, it binds to the conserved C-terminus of FtsZ and was suggested to compete with FtsA and ZipA to inhibit FtsZ ring formation. (*Hu and Lutkenhaus, 2000*; *Shen and Lutkenhaus, 2009*). Note, that FtsZ-mts comprises amino acids 1–366 of *E. coli* FtsZ, but instead of the C-terminally conserved tail, it comprises YFP and a membrane targeting sequence. The proposed interference of the C-terminal domain with FtsZ is therefore not involved in the system described above, which suggest that the inhibitory activity of the C-terminal domain is not required in a minimal system to inhibit FtsZ at compartment poles. In the context of a living cell, the inhibitory activity of the C-terminal

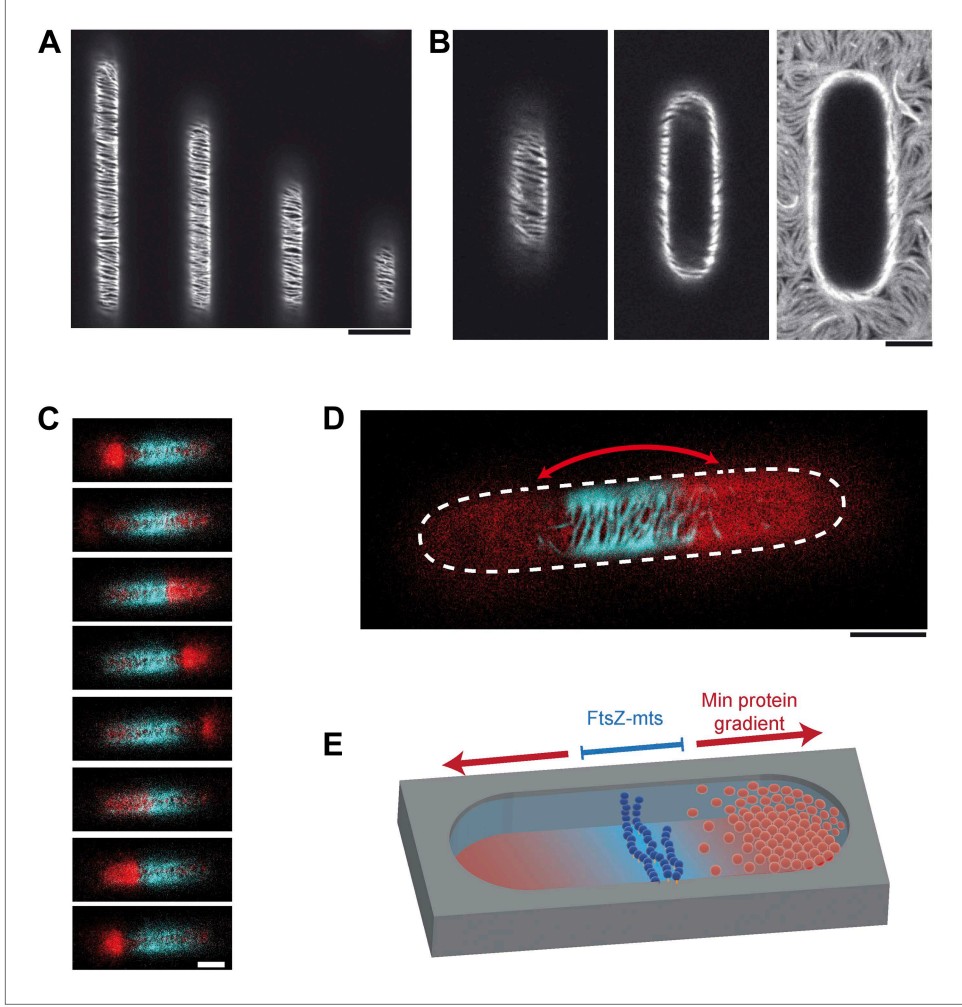

**Figure 2**. Reconstituted protein gradients coordinate localization of downstream targets in cell-shaped compartments. (**A**) Confocal image of the tubulin homolog FtsZ-mts on the bottom of membrane-clad compartments. FtsZ-mts assembles into bundles which are aligned perpendicular to the length axis of the compartments. Scale bar: 10 µm. (**B**) Images with the focal plane at the bottom, middle and top of the compartment, respectively. The buffer was not yet reduced to a level below the upper level of the chip. Therefore membrane and FtsZ-mts network are still intact on the upper level of the chip. Scale bar: 5 µm. (**C** and **D**) When the gradient forming Min system (red: MinE.Atto655) and FtsZ-mts (blue) are reconstituted together, the Min gradient coordinates the localization of FtsZ-mts to the middle of the compartments. Scale bar: 5 µm. (**C**) Time between frames: 20 s. Negative controls in *Figure 2—figure supplement 1* confirm that MinD and MinE are required to form dynamic patterns and that MinC is needed in addition to deplete FtsZ-mts from the membrane. (**E**) Schematic image of reconstituted FtsZ-mts positioning.

The following figure supplement is available for figure 2:

**Figure supplement 1**. MinD MinE and ATP are required to form dynamic patterns and MinC is needed to deplete FtsZ from the membrane.

domain of MinC might provide an additional mechanism for disrupting FtsZ, rendering the Min system more robust.

The N-terminal domain of MinC inhibits FtsZ polymerization by binding to an alpha-helix of FtsZ at the interface of FtsZ subunits in a filament (*Hu et al., 1999*; *Shen and Lutkenhaus, 2010*). Since the FtsZ-binding region for the C-terminal domain of MinC is not present in FtsZ-mts, the major inhibitory activity contributing to FtsZ-inhibition in our synthetic system should originate from the N-terminal domain of MinC. To test this hypothesis, we used a MinC mutant (MinC-G10D) with lower inhibitory

activity of its N-terminal domain (*Hu et al., 1999*; *Shen and Lutkenhaus, 2009*, *2010*). The MinC-G10D mutant has reduced ability to inhibit FtsZ in vivo and in vitro, but interacts with MinD (*Hu et al., 1999*). We first characterized MinC-G10D by co-reconstituting it at different concentrations with FtsZ-mts, MinD and fluorescently labeled MinE on flat supported membranes. Consistent with the behavior of wild type MinC, we found that the self-organizing Min protein patterns were unperturbed at low concentrations of MinC-G10D, and that the Min patterns were weakened by high concentrations of MinC-G10D (*Figure 3A*). The weakening of Min patterns for high concentrations of MinC is likely due to an overlap of the binding sites for MinC and MinE on MinD (*Ma et al., 2004*; *Wu et al., 2011*), which might result in a competition of MinC with MinE for binding sites on MinD at large MinC concentrations. The weakening of Min Protein patterns therefore suggests that MinC-G10D binds to MinD. However, in contrast to MinC the MinC-G10D mutant was inefficient in depolymerizing FtsZ-mts on flat membranes, as well as at the poles, when co-reconstituted in microcompartments (*Figure 3B,C*, *Figure 3—figure supplement 1*). These observations confirm that the inhibitory activity in the synthetic system originates from the N-terminal domain of MinC and that the inhibitory activity of the N-terminal domain is sufficient to inhibit FtsZ-polymerization at compartment poles in the context of a minimal system.

## Establishment of protein gradients is dependent on compartment geometry

### Gradient formation and FtsZ localization dependent on compartment length

Experiments with aberrantly shaped cells, such as round and filamentous *E. coli*, have shown that dynamic pattern formation depends on compartment geometry. However, the mode of pole-to-pole oscillations should be stable over a substantial variation in length. As cells grow and double their length before division, stable pole-to-pole oscillations of the Min proteins need to be established over a spatial range of at least two times the length of the daughter cells. We therefore hypothesized that stable pole-to-pole oscillations should be found in microstructures of different length, where the range should vary at least by a factor of two, if this length-dependent robustness is intrinsic to the Min system. To test this hypothesis, we reconstituted the oscillations of MinD and MinE in microcompartments of systematically varying length from 12 µm to 245 µm and investigated the stability of Min patterns at different compartment lengths. In compartments with a constant width of 10 µm, the Min system oscillated along the long axis of the compartments (*Zieske and Schwille, 2013*) (*Video 1*). We observed oscillations with multiple minima and maxima in long compartments, and found that every oscillation mode occurred at a defined range of the compartment length (*Figure 4A*). In most of the short compartments with an aspect ratio (length to width) below 4.5, we observed pole-to-pole oscillations, in rare cases they prevailed up to aspect ratios of 5.5.

In particular, these experiments demonstrate that pole-to-pole oscillations are stable (i.e., no other oscillation mode occurs in those compartments) over a significant length range of 15 µm–35 µm. In other words, although the length of these cavities varies by about a factor of 2.3, the oscillatory behaviour is conserved. This result shows that no auxiliary proteins besides the MinDE protein system are required to adopt the Min wavelength to the varying length of a growing bacterium, demonstrating the power of this system as a robust spatial cue.

In the shortest of these compartments, with an aspect ratio of 1.2, we occasionally observed an aberration of the oscillation axis from the long axis (*Video 2*), which is indicative of a transition from stable pole to-pole oscillations to the patterns observed in round compartments and cells (*Zieske and Schwille, 2013*). *Figure 4A* provides an overview of the oscillation modes with respect to the length-to-width ratio of the compartment.

While the length dependent increase of oscillation modes was well conserved in our experiments, the oscillation period was prone to small perturbation. In particular, we observed a slight increase of the oscillation period over time, which might be a result of buffer evaporation (*Figure 4—figure supplement 1*). Thus, oscillation periods between less than a minute and 4 min were measured. This time scale is well in agreement with measurements of 0.5–2 min for the oscillation period in living cells (*Raskin and de Boer, 1999b*).

The strong fluctuation of oscillation periods in living cells, as well as between different samples of the synthetic system, render a systematic characterization of cell length dependent protein oscillations challenging. To overcome these problems and to better characterize the dependence on compartment length, we acquired time-lapse movies, capturing numerous compartments simultaneously. Then, we

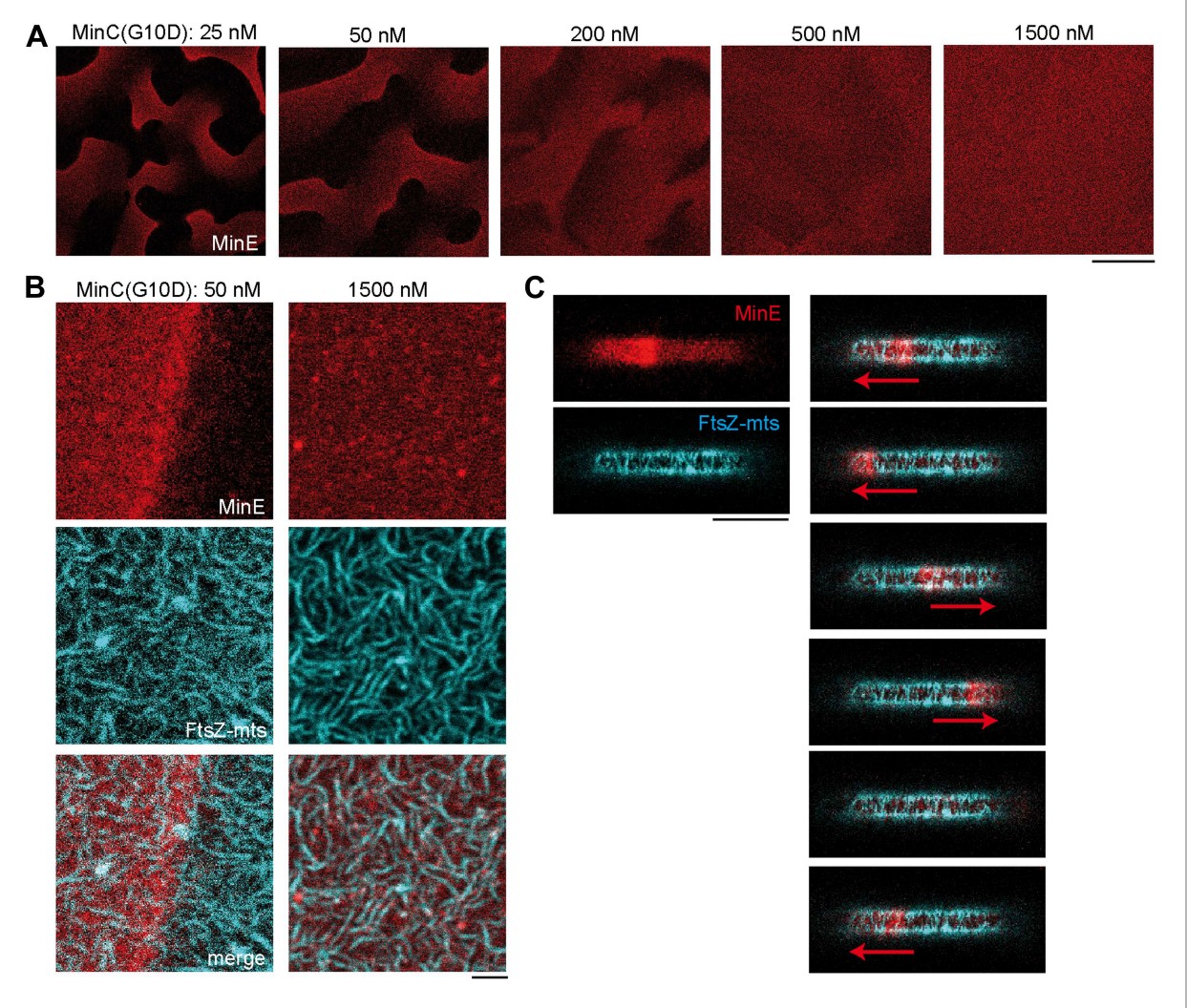

**Figure 3**. MinC-G10D does not inhibit polymerization of FtsZ-mts. (**A**–**C**) FtsZ-mts (blue), MinD, MinE (5% labeled with Atto655: red) were reconstituted with the MinC-G10D mutant. Concentrations of MinD, MinE and FtsZ-mts are constant for all images. (**A**) Confocal images of Min protein pattern (MinE.Atto655: red) on flat supported membranes at different concentrations of MinC-G10D. High concentrations of MinC-G10D disturb Min protein patterns. The contrast between images is not comparable and is increased for higher concentrations of MinC-G10D, because the MinE intensity at the membrane decreased with higher concentrations of MinC-G10D. Scale bar: 50 µm. (**B**) Confocal images of the Min system (MinE.Atto655:red) and FtsZ-mts (blue) on flat membranes at different concentrations of MinC-G10D demonstrate that MinC-G10D is inefficient in disturbing FtsZ-mts networks. Scale bar: 2 µm. (**C**, *Figure 3—figure supplement 1*) In cell-shaped micro compartments with MinD, MinE, FtsZ-mts and 50 nM MinC-G10D the Min system (MinE.Atto655: red) oscillates from pole-to-pole. However MinC-G10D does not inhibit polymerization of FtsZ-mts (blue) at the compartment poles. Time between frames: 90 s, scale bar: 10 µm.

The following figure supplement is available for figure 3:

**Figure supplement 1**. MinC-G10D does not inhibit polymerization of FtsZ-mts at compartment poles.

compared Min oscillations in micro compartments with different aspect ratios up to 5.5, which were acquired at the same time on the same chip. Imaging multiple compartments on the same chip allowed us to analyse Min oscillations in about 200 compartments under the same experimental conditions at the same time. We found that the oscillation period between aspect ratios of 1.5 and 5.5 increased from 79 s to about 165 s. For higher aspect ratios we only observed higher oscillation modes. For very low aspect ratios of 1.2 the mean value for the oscillation period was slightly higher than for 1.5 (*Figure 4B*). Note that in very small cells a stochastic switching of Min protein pattern between the poles have been observed (*Fischer-Friedrich et al., 2010*) and that a longer oscillation period for very

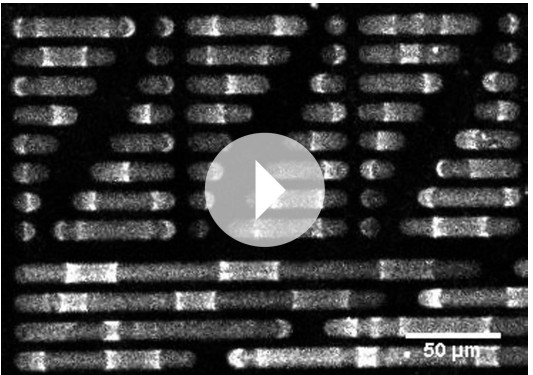

**Video 1**. Reconstituted Min protein oscillations in microcompartments of different length. This video shows a confocal time lapse movie of reconstituted MinD/MinE pattern in cell-shaped micro-compartments. 10% MinE are labeled with Alexa 488. The movie follows the dynamics of Min protein pattern for 17 min.

small compartments might represent a transition from normal pole-to-pole oscillation to different dynamics in very small compartments.

To address how downstream processes of the Min gradient are affected by the length dependent oscillation modes in compartments of different length, we investigated FtsZ-mts localization in these compartments. We found that higher oscillation modes resulted in multiple minima of the MinC profile and consequently in multiple distinct regions of FtsZ-mts assembly (*Figure 5*).

## Gradient formation dependent on compartment width

How does the cell diameter affect cell organization, and in particular gradient formation? Tools to answer this question for the chosen system in vivo are limited, because in living *E.coli* bacteria it is challenging to systematically modify the width of a cell. Thus, it is unknown whether the constant diameter of cells plays a significant role as a geometric cue for septum positioning. Compared to living cells, our in vitro system is much more flexible in pre-setting a defined geometry and reaction volume, enabling us to address for the first time the role of cell width for Min gradient formation and thus, as a geometrical cue for cell organization. In compartments with a small width of 10 μm, only oscillations along the long axis were supported (*Video 1*, *Figure 6A*). Interestingly however, larger widths also resulted in oscillations along the short axis of a compartment (*Figure 6B*, *Video 3*). Even larger widths resulted in more complex oscillation patterns (*Figure 6—figure supplement 1*, *Video 4*). These experiments revealed that a fixed small width of the cell is important for Min gradient formation along the long axis.

## Geometry modulated partitioning of min proteins during septum closure

A dramatic local change in cell width occurs during every cell division in *E. coli* in the middle of the cell due to septum closure. Theoretical modeling suggested that this change in geometry might account for an equal partitioning of Min proteins to both daughter cells. (*Di Ventura and Sourjik, 2011*) In other words, septum constriction is hypothesized to provide a geometrical cue to switch from the pole-to-pole oscillations of Min proteins to a symmetric double oscillation, which enables the daughter cells to trap equal amounts of Min proteins. In live cells it is challenging to directly observe the partitioning of proteins, because the constricting septum of *E. coli* becomes smaller than the optical resolution limit during the proposed process. At the same time, the Min protein system is a highly dynamic system, rendering time-lapse imaging with current superresolution techniques difficult. To experimentally test the proposed partitioning of Min proteins during septum constriction, compartments that mimicked morphologically different stages of bacterial cell division were designed. Using cell-shaped compartments with a systematically varying width of the artificial septum, we analyzed whether different patterns occurred in more 'constricted' compartments, as compared to 'non-constricted' compartments.

We found that double oscillations occurred indeed in compartments with a small neck in the middle of the compartment. As a negative control, normal pole-to-pole oscillations were sustained in compartments with the same length, but constant cell width. Finally, geometries mimicking the full septum closure, that is, separation in two cells, induced independent pole-to-pole oscillations in each 'daughter' compartment (*Figure 6C*, *Figure 6—figure supplement 2 and 3*, *Video 5*). This finding presents the first direct experimental evidence that equal Min protein distribution during cell division is a result of geometrical cues. Furthermore, it strikingly confirms theoretical modeling predictions, where septum constriction supports partitioning of the Min proteins (*Di Ventura and Sourjik, 2011*).

To address the role of cell length for equal partitioning of Min proteins during septum closure, we performed similar experiments using cell shaped compartments with different length. We demonstrated

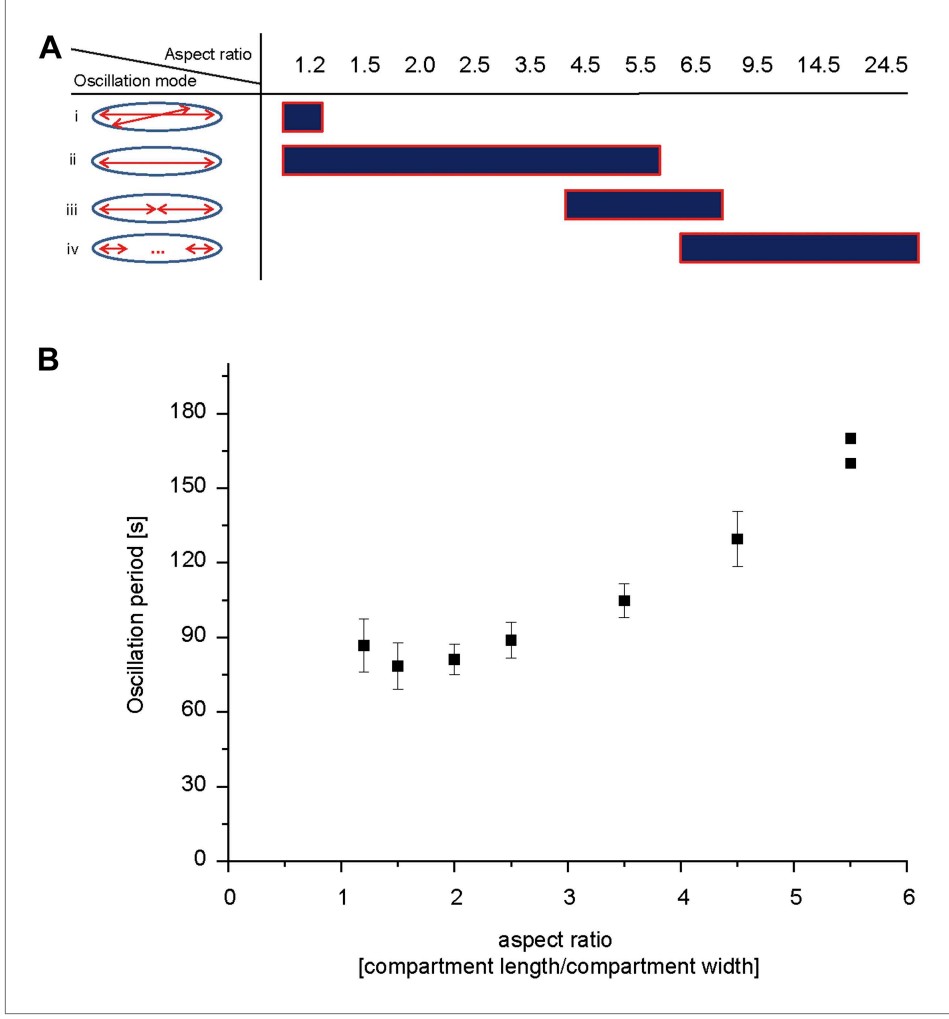

**Figure 4**. Compartment length dependent protein oscillations. (**A**) Experiments such as in *Video 1* were analyzed for the oscillation mode in the corresponding compartments and categorized as (i) oscillations with aberration from the long axis of the compartment, (ii) pole-to-pole oscillations, (iii) double oscillations, (iv) higher oscillation mode (at least triple oscillations). (**B**) Oscillation period of pole to pole oscillations in dependence of compartment length. Simultaneous time-lapse acquisition of Min protein patterns in multiple compartments allowed highly comparable experimental conditions. In total, 27 to 29 compartments of seven different aspect ratios (1.2, 1.5, 2.0, 2.5, 3.5, 4.5, 5.5) on the same time lapse movie were analyzed. Out of 27 compartments with an aspect ratio of 1.2, one compartment harboured oscillation that deviated from the length axis. Out of 28 compartments with an aspect ratio of 4.5 we observed 16 compartments, and out of 29 compartments with an aspect ratio of 5.5 we observed 27 compartments with double oscillations or oscillations which switched between pole-to-pole oscillations and double oscillations. All remaining compartments harboured pole-to-pole oscillations for which the mean oscillation period and standard deviation was calculated. For an aspect ratio of 5.5 the two data points are directly plotted, due to the small sample size. *Figure 4—figure supplement 1* demonstrates that the oscillation period increases over time.

The following figure supplement is available for figure 4:

**Figure supplement 1**. Kymograph of MinE oscillations.

that a switch from asymmetric pole-to-pole oscillations to symmetric double oscillations occurred at a certain cell length. However, in longer compartments, double oscillations occurred already in unconstricted cells und thus, a constriction in the middle of the cell had no additional effect on pattern formation. In very short cells, on the other hand, a constriction in the middle of the compartment resulted in a new length axis of the two compartments perpendicular to the length axis of the unconstricted

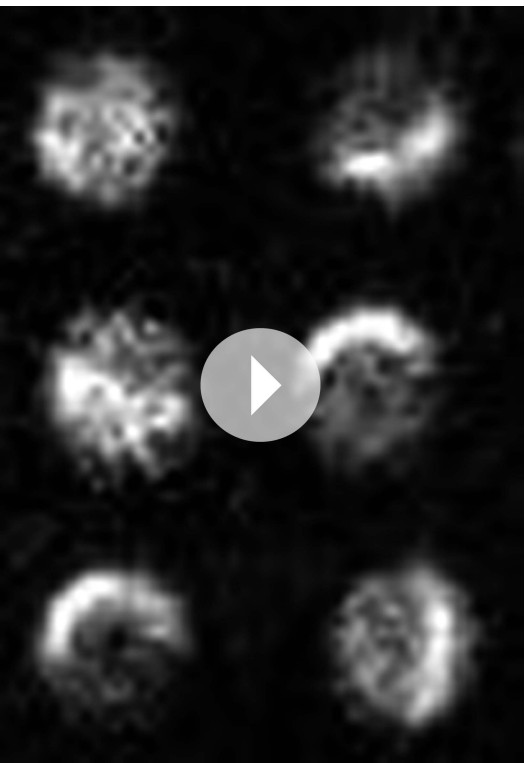

**Video 2**. Min protein pattern in short compartments. This movie shows confocal time-lapse movies of a MinD and MinE patterns (doped with 10% MinE.Alexa488) in 12 μm long and 10 μm wide compartments. Six time-lapse movies with compartments in which the Min patterns aberrate from a stable pole-to-pole axis are stitched together. The protein patterns are followed over a time of about 17 min.

'mother' compartment (*Figure 6D*). Thus, a re-orientation of the pattern occurred during constriction, from oscillations along the long axis in the unconstricted compartment, to perpendicular new length axes. These findings demonstrate that an optimal range of cell length exists were Min gradients represent an efficient cue for septum formation. In addition, the Min partitioning to both daughter cells can be controlled by changing the local compartment width during septum closure in the middle of the cell.

## Discussion

We demonstrated that coordinated spatial control by intracellular protein gradients can be reconstituted in cell-free systems, and that these gradients are modulated by geometric parameters. The reconstitution of self-organizing protein systems in membrane clad soft-polymer compartments represents a simple approach to study membrane interacting proteins in a defined sample volume and geometry. Applying this system, we particularly tackled three major questions. First, what are the minimal requirements for protein gradient formation? Second, can the organized localization to predefined sites be achieved by reading out the positional information of a minimal protein gradient system? And finally, what are the geometric boundary conditions to generate and modulate protein gradients?

To address the minimal requirements for gradient formation, we considered the Min protein system as a prototype for a gradient forming system and accomplished the reconstitution of a time-averaged nonhomogeneous concentration profile with the lowest concentration in the middle of the compartment. We demonstrated that the minimal requirements for Min gradient formation comprised only a negatively charged membrane, a cell-shaped compartment, ATP as an energy source, and the two antagonistic proteins MinD and MinE.

To investigate whether the reconstituted gradient provides an efficient cue to organize cellular space by positioning down-stream targets, we co-reconstituted the MinD/E system with the cell division protein FtsZ-mts and the FtsZ inhibitor MinC. This cell-free bottom-up system recapitulated remarkably cell-like properties, such as coordinated spatial coordinated localization of FtsZ-mts to the middle of a cell-like compartment. Interestingly, although it is proposed that MinC has two inhibitory domains which act on FtsZ–FtsZ interactions and on the ternary system FtsZ-FtsA/ZipA, respectively (*Shen and Lutkenhaus, 2009, 2010*), our data suggest that MinC can already inhibit FtsZ-localization at compartment poles by only disrupting FtsZ–FtsZ interactions. While we do not exclude that the interaction of MinC with the conserved C-terminal domain, which interacts with FtsA and ZipA, is a parallel mechanism to inhibit FtsZ polymerization at cell poles and possibly make FtsZ structures more sensitive to the action of the Min system, our results show that the minimal biochemical requirements to displace FtsZ from compartment poles by the Min system do not involve the disruption of FtsZ-membrane binding sites.

Moreover, we demonstrated that very long compartments harbored a MinC protein concentration profile with multiple minima. While a normal growing wild type cells typically comprises a Min concentration gradient with only one minimum for division protein assembly, multiple minima resulting in more than one sides for cell division, are potentially advantageous in filamentous cells. If transient unfavorable environmental conditions inhibit cell division but not cell elongation, divisions at multiple

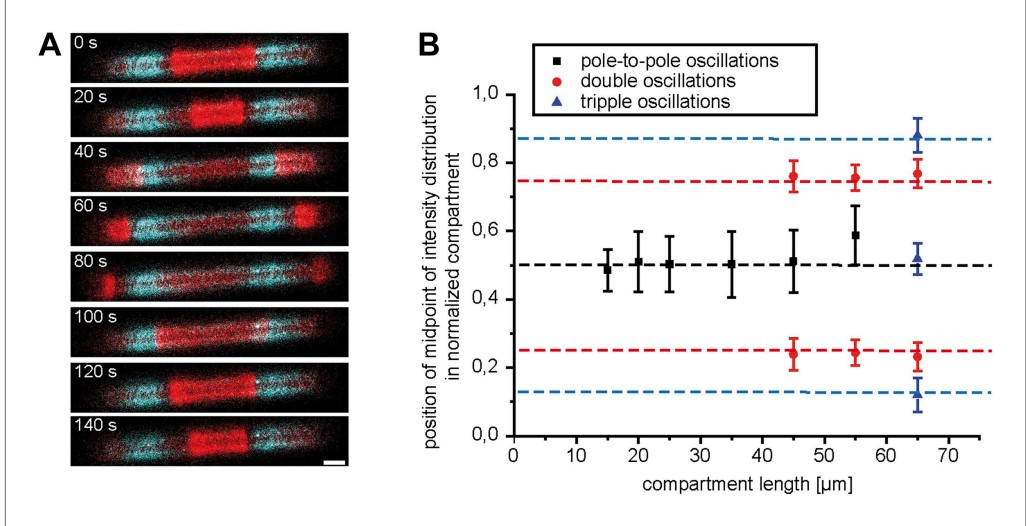

**Figure 5**. Compartment length affects FtsZ localization. (**A**) In compartments with double oscillations of the Min system (red: MinE.Atto655), FtsZ-mts (blue) localizes to two regions along the cell length. Scale bar: 5 µm. (**B**) Localization of FtsZ-mts along the length axis of an artificial cell for pole-to-pole oscillations (black), double oscillations (red) and triple oscillations (blue) in comparison to expected values (dotted lines), considering the symmetries of oscillation modes of Min protein along the long axis of a compartment. For pole-to-pole oscillations in 55 µm long compartments: n = 3 (For 55 µm the majority of structures revealed double oscillations and only in a minor fraction of these long structured occured pole-to-pole oscillations). For all other values, n >9.

sides could ensure the re-establishment of a normal cell length when conditions, which support division, reoccur.

To systematically address how compartment geometry modulates Min protein gradients, we reconstituted Min protein patterns in compartments with systematically varying shape. Varying the length of the compartments, we found that robust pole-to-pole oscillations occur in compartments, even if the length of these compartments was varied by a factor of 2.3. In a live cell, this robustness of the Min oscillations with respect to cell length should be highly significant, because the length of living cells is not constant over time, and during the life cycle of *E. coli*, the cell doubles in length. The observation that the Min protein patterns in vitro deviate from a stable axis in very short compartments, and that higher order oscillations systematically occur the longer the compartment is, confirm that the MinD/E patterns in synthetic compartments are strikingly similar to the Min pattern of round, wild type and filamentous cells in vivo (*Fu et al., 2001*; *Hale et al., 2001*; *Corbin et al., 2002*; *Shih et al., 2005*; *Raskin and de Boer, 1999b*).

Interestingly, when we increased the width of the compartments, the Min proteins oscillated along the short axis of the compartment. The cause of oscillations switching along the length axis to oscillations along the short axis is not known. However, the width at which this switching occurs might be correlated with the wavelength of the Min protein patterns and therefore related to the mechanism for determining the wavelength of the Min system. It is also unknown whether a similar gradient along the short axis of the cell is used in living cells, and which factors might contribute on the bacterial scale to regulate gradients along the short axis.

However, note that FtsZ involving cell-division in certain bacteria, such as *Laxus oneistus*, occurs along the long axis. (*Polz et al., 1992*; *Leisch et al., 2012*) Although the spatial cues to determine the position of their division plane are unknown, it is intriguing to speculate that gradients along the short axis might also provide an effective cue to spatially organize space along the short axis Thus, protein gradients might also orient the cell division plane in cells which divide along the length axis of the cell.

In line with our in vitro results, Min oscillations along the short axis of a bacterial cell might be accomplished by an increased width as compared to an *E.coli* cell, or by smaller scales of Min protein patterns through modified reaction and/or diffusion rates of the proteins. However, also other systems which generate gradients along the length axis of a bacterial cell, such as activity gradients of

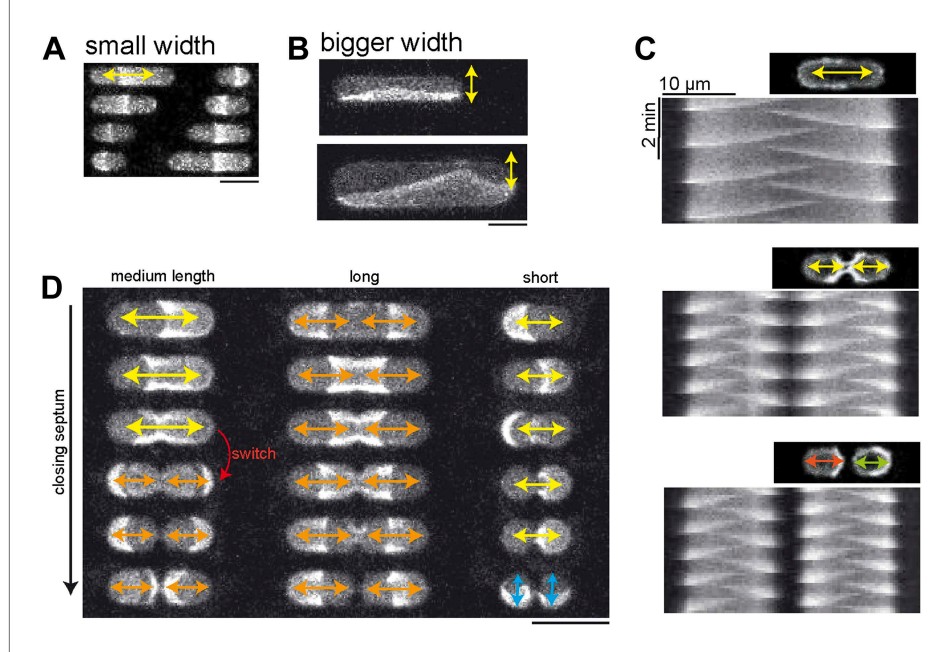

**Figure 6**. Compartment width affects gradient formation. (**A**) Pole to pole oscillations occur is narrow compartments. Scale bar: 20 μm. (**B**) In wider compartments the proteins oscillate parallel to the length axis of the compartment. Scale bar: 20 μm. In compartments with an even larger width more complex patterns occur as depicted in *Figure 6—figure supplement 1*. (**C**) Confocal images and kymographs of MinE pattern in compartments with different degrees of 'constriction' depict pole-to-pole oscillation in compartments with a constant width (upper panel). In compartments with a narrow width in the middle of the compartment double ascillations occur. In separated compartments independent pole-to-pole oscillations are observed. (**D**) Oscillation modes in compartments with different lengths and stepwise narrowing width at the middle of the compartment. Left: Compartments without constriction or only slight decrease of the width in the middle harbor pole-to-pole oscillations (asymmetric oscillations, yellow arrows). Compartments with a narrow width in the middle harbor double oscillations (symmetric oscillations, orange arrows). A series of time-lapse images as well as kymographs of the third and fourth compartment, in which the transition from pole-to-pole to double oscillation occurs are shown in *Figure 6—figure supplement 2*. *Video 5* shows the dynamic pattern of the third and fourth compartment in this image. Middle: In longer compartments double-oscillations occur due to the increased length, even if the width along the length of the compartment is constant. Right: In very short cells the constriction of the compartments results in two connected compartments, with length axis perpendicular to the length axis in non-constricting compartments. Thus the Min proteins oscillate along the new length axis, which is perpendicular to the oscillation axis in non-constricting short compartments (blue arrows). 1 μM MinE (doped with 10% MinE.Atto655), 1 μM MinD, Scale bar: 20 μm *Figure 6—figure supplement 3* demonstrates that oscillations happen between each 'constricting septum' if the compartment harbors two septa.

The following figure supplements are available for figure 6:

**Figure supplement 1**. Min proteins self-assemble into complex pattern in large compartments.

**Figure supplement 2**. Min proteins patterns switch from asymmetric pole-to-pole oscillations to symmetric double oscillation when the neck in the middle of a compartment decreases.

**Figure supplement 3**. Min oscillations in compartments with multiple septa.

phosphorylated proteins (*Chen et al., 2011*), might provide mechanisms to generate gradients along the short axis.

In addition to varying the overall width of a compartment, we mimicked septum closure in the middle of the compartments and thereby simulated morphologically different stages of bacterial cell division. While it has been controversial how the pool of Min proteins is distributed to the two daughter cells after cell division, we provide experimental evidence for the theoretical model that Min

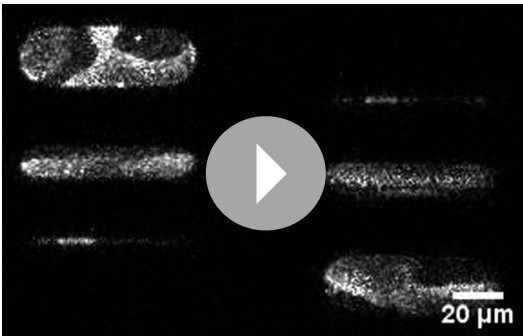

**Video 3**. Compartment width dependent orientation of Min oscillations. This movie demonstrates how the oscillation axis switches depending on the width of the sample compartment. The confocal time-lapse movies of MinD and MinE patterns (doped with 10% MinE. Atto655) follows the dynamic oscillations over a time of 6 min.

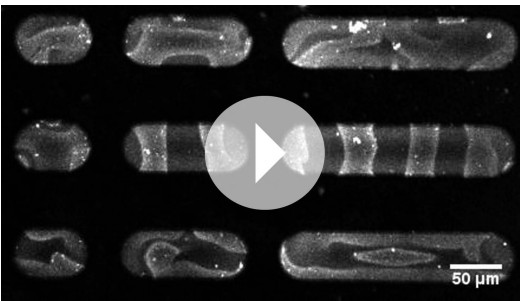

**Video 4**. Complex Min protein pattern in large compartments. This movie shows a confocal time-lapse movie of a MinD and MinE patterns (doped with 10% MinE.Alexa488) in 50 μm wide compartments. The protein patterns are followed over a time of 6 min.

protein partitioning is a result of geometric variation during septum constriction (*Di Ventura and Sourjik, 2011*).

In summary, we reconstituted a cell-free bottom-up system which recapitulates cell-like properties, such as coordinated spatial control for division site placement by intracellular protein gradients. In particular, our results unravel the minimal requirements for localization of the cell division protein FtsZ. Moreover, our findings demonstrate that the interplay of geometrical cell boundaries and self-organizing Min proteins determine the orientation of Min gradients, the number of potential FtsZ assembly sites, as well as Min protein partitioning during cell division. Thus, our synthetic system opens a way to study complex organization principles in a simplified environment, and provides novel insights into the basic biophysical and biochemical requirements for gradient formation.

## Material and methods

### PDMS micro-compartments

Using photolithographic techniques, resist micro-patterns were produced on Si wafers mainly as described in reference (*Zieske and Schwille, 2013*). Photoresist patterns (ma-P 1275, micro resist technology GmbH, Germany) of about 10 μm height on top of Si wafers (Si-Mat, Kaufering, Germany) were produced by photolithography. The chrome mask was purchased from Compugraphics Jena GmbH. To better mimic the curvature of the cellular membrane, we left a small gap between photoresist and mask during exposure of the mask to UV light, resulting in more tilted walls as compared to the contact mode. For reconstitution of Min protein oscillations along the length axis of a compartment, the mask patterns had a width of 10 μm. The Si wafers were coated with chlorotrimethylsilane (Sigma–Aldrich, St. Louis, MO) to prevent sticking of PDMS to the wafer. PDMS (Sylgard184, Dow Corning, Midland, MI) was mixed at a ratio (monomer to cross-linker) of 10:1 and degased under vacuum. The PDMS mixture was then poured on top of the wafer. Standard glass coverslides where then manually pressed into the liquid PDMS on top of the wafer, leaving a PDMS layer of about 30 μm between the wafer and the glass coverslide. After curing the PDMS at 80°C overnight, the bulk PDMS layer was peeled off. With the help of a razor blade, the glass coverslide with the about 30 μm thin, micro structured PDMS layer was carefully separated from the wafer and used as sample support (*Figure 1—figure supplement 2*). The microstructured PDMS/glass devices were stored at room temperature until further use. Before the microstructured PDMS was used as a membrane support, it was sonicated for 5 min in ethanol, washed with water, air dried, and treated with air plasma.

### Design of microcompartments

The two-dimensional geometry at the upper level of the compartment is determined by the patterns on a chrome mask. The profiles of the micro compartments can be designed with a flat bottom or a tapered bottom (*Figure 1—figure supplement 1C*). The Min protein oscillations and Min gradient formation were supported by both profiles. However, the image quality was better when the bottom of the compartments was flat. The advantage of a tapered bottom was that these structures allowed

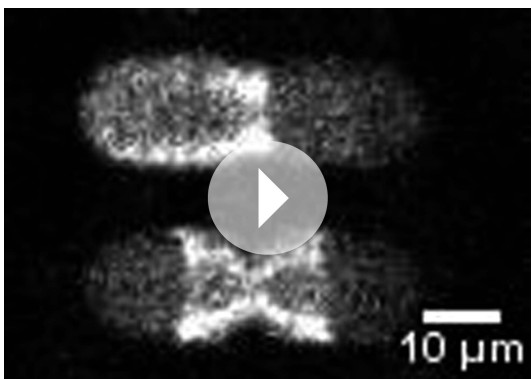

**Video 5**. Min protein pattern in compartments with a small neck in the middle of the compartment. This movie shows a confocal time-lapse movie of a MinD and MinE patterns (doped with 10% MinE.ATTO655) in compartments with a small neck in the middle. The width of the neck in the lower compartment is smaller as compared to the upper compartments and induces a switch from pole-to-pole oscillation to double oscillations. Time of the movie: 2 min.

FtsZ-mts to align perpendicularly to the length axis of the compartments (*Figure 1—figure supplement 1D*). Thus, all compartments containing FtsZ-mts (*Figures 2,3,5*) had a tapered bottom. To keep the boundary conditions comparable, the MinCDE oscillations and gradients in *Figure 1* were reconstituted in tapered compartments as well.

To better visualize the oscillation mode and measure the oscillation period of MinD/E pattern, compartments with a flat bottom were employed (*Figure 4*, *Video 1 and 2*). Note that the oscillation mode in flat compartments (*Figure 4A*) is in agreement with FtsZ localization in tapered compartments (*Figure 5B*). Compartments with constrictions and compartments for determining width dependent Min patterns had a flat bottom, as well (*Figure 6*, *Video 3–5*).

Experiments with MinD/E pattern in compartments with constricting septum (*Figure 6* and *Figure 6—figure supplement 1–3*): The walls of the compartments were slightly tilted and the bottom of the compartment was flat. The width at the unconstricted region was about 12 μm at the top and 10 μm at the bottom of the compartments. The septal region of compartments, in which the switch of the oscillation modes occurred, had a width of about 2 μm at the top. The resolution limit in z-direction and the tilted PDMS-surface through which the laser beam had to pass, render an exact description of the profile at the septum challenging, However a confocal x/z-scan of the septum is depicted in *Figure 6—figure supplement 3C*. Also note that a meniscus due to the surface tension of the buffer might render the actual cross-sectional area of the buffer smaller than the profile of the structure suggests.

## Supported lipid membranes

*E. coli* lipid membrane within the micro compartment was prepared as described previously (*Zieske and Schwille, 2013*). Two-dimensional motility of the lipids within the membranes was confirmed by labelling the membrane with 0.1% DiI (FAST DiI, Invitrogen, Carlsbad, CA) and performing FRAP experiments. DiI labeled membranes were also applied to acquire compartment profiles.

## Protein reconstitution assay

Proteins, ATP and/or GTP (depending on experiment) of defined concentration were added to a buffer reservoir of 200 μl on top of membrane clad supports. Dynamic Min patterns and/or FtsZ-mts bundles formed by self-organization on top of the membrane. At this step the Min patterns did not oscillate but formed travelling waves. Afterwards the buffer above the microstructures was removed by pipetting. Restricted to a small buffer compartments within the micro structures the Min protein patterns start to oscillate in cell-shaped geometries. To limit evaporation a lid was placed on top of the plastic ring surrounding numerous microstructures. The top of individual chambers was open with a buffer/air interface (*Figure 1—figure supplement 2*).

## Proteins

MinC, eGFP-MinC, MinD, eGFP-MinD and MinE were purified with N-terminal His-tags as previously described (*Loose et al., 2008*, *2011*). Purification of FtsZ-mts was originally described by the Erickson-group (*Osawa et al., 2008*). MinE was labeled with Alexa Fluor 488 C5 Maleimide (Molecular Probes, Carlsbad, CA) or ATTO655 Maleimide (ATTO-TEC, Siegen, Germany) according to the manufactures manual.

The plasmid for overexpression of MinC codes for an open reading frame for MinC connected to an N-terminal hexahistidine tag by a linker (*Loose et al., 2011*). The MinC-G10D mutant was obtained by site-directed mutagenesis (QuikChange II, Agilent Technologies, Santa Clara, CA) of this plasmid to obtain pET28a-MinC(G10D).

MinE without membrane targeting sequence (MinE(Δ3–8)) was generated by deleting the coding sequence for aminoacids 3–8 of MinE of the overexpression plasmid for MinE (*Loose et al., 2008*) using a 'GeneArt, Seamless Cloning and Assembly Enzyme Mix' (Invitrogen, Life Technologies, Carlsbad, CA) and primers TTCCGCGATGCGAATTCGGATCCGCGACC and AATTCGCATCGCGGAAGAAAAACACAGCCAACA to obtain pET28a-MinE(Δ3–8). MinC-G10D and MinE(Δ3–8) were purified like MinE. Purified MinE(Δ3–8) was labeled with Alexa647 maleimide (Molecular Probes, Carlsbad, CA) according to the manufactures manual.

The sample buffer in all experiments contained 25 mM Tris–HCl pH 7.5, 150 mM KCl, 5–15 mM MgCl$_2$. (5 mM MgCl$_2$ if only the MinD and MinE were reconstituted, 15 mM MgCl$_2$ if FtsZ-mts was reconstituted) When Min proteins where included in the experiments, the buffer was supplemented with 2.5 mM ATP. When FtsZ-mts was used 3 mM GTP was added to the buffer. Protein concentrations used in this study were as follows. MinC or eGFP-MinC: 0.05 µM, MinE: 1 µM, MinD: 1 µM (depending on the experiment MinD was supplemented with 10% eGFP-MinD), FtsZ-mts: 1 µM. When MinE was imaged 10% of MinE were substituted with the labeled version of MinE.

Note that these concentrations refer to the average concentrations in the sample buffer before the buffer volume was reduced below the upper level of the micro compartments.

## Microscopy

Microscopy: Image acquisition was performed on a ZEISS (Jena, Germany) LSM780 confocal laser scanning microscope (inverted microscope) equipped with a ZEISS Plan-APO 25×/NA 0.8 objective and ZeissC-Apochromat 40×/1.20 objective.

For acquiring images of Min pattern and FtsZ bundles the focal plane was typically adjusted to the bottom of the compartments (if not stated otherwise). Note that the bottom area of the microstructures is smaller than the top area. Therefore the sizes of structures on the pictures might appear smaller than indicated in the text (text always refers to upper rim of microstructures).

The relative intensity profiles along the length axis of compartments can be more accurately determined if the focus plane is in the middle of the compartment, because at this plane the relative intensity profiles at different time steps are less prone to errors due to slight focal shift. Thus, data in *Figure 1D* and *Figure 1—figure supplement 3* were extracted from time-lapse images with the focal plane in the middle of the compartments.

## Acknowledgements

We thank Michaela Schaper for help with cloning, Senthil Arumugam for discussions and providing FtsZ-mts and Martin Loose for comments on previous versions of the manuscript. KZ is supported by 'The International Max Planck Research School for Molecular and Cellular Life Sciences' (IMPRS-LS). KZ and PS are associated with CeNS (Center for nanoscience) of the Ludwig-Maximilians University Munich.

## Additional information

### Funding

| Funder | Grant reference number | Author |
| --- | --- | --- |
| Deutsche Forschungsgemeinschaft | Collaborative Research Centre 1032 | Petra Schwille |

The funder had no role in study design, data collection and interpretation, or the decision to submit the work for publication.

### Author contributions

KZ, Conception and design, Acquisition of data, Analysis and interpretation of data, Drafting or revising the article; PS, Conception and design, Drafting or revising the article

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
