## [Decision Letter]

Thank you for sending your work entitled “Reconstitution of self-organizing protein gradients as spatial cues in cell-free systems” for consideration at *eLife*. Your article has been favorably evaluated by Richard Losick (Senior editor) and 3 reviewers, one of whom is a member of our Board of Reviewing Editors.

The Reviewing editor and the other reviewers discussed their comments before we reached this decision, and the Reviewing editor has assembled the following comments to help you prepare a revised submission.

As you will see the referees have raised a number of points, a large number of which pertain to providing more details of experimental details and to some additional controls. However, one major concern that we would like you to further address is the role of MinC in FtsZ ring positioning in your studies. For example, is it possible to assemble more focused rings if full length MinC were used? The experiments should also be performed with the MinC-G10D mutant to correlate the results of the *in vitro* experiments with *in vivo* experiments. We would like you to consider and respond to the other comments made by the referees, if possible with experiments or with explanations to rebut the points raised. With the appropriate changes, your work will make a very important contribution to the fields of bacterial cell biology and cytokinesis in general.

*Reviewer 1*:

The authors take a bottom up approach to reconstitute the oscillatory motion of the MinCDE system. They also present evidence demonstrating the sufficiency of the system in positioning FtsZ in the medial region within cylindrical compartments. They also show that while the Min gradient is independent of the chemical nature of the lipid bilayer, geometric cues play a rather critical role in its patterning. The work is major achievement in reconstituting minimal systems. This also provides major insights into determinants of spatial organization in biological systems.

MinDE oscillations are shown in Figure 1. While it’s clear that MinD and MinE are sufficient to set the oscillations and achieve the maximum concentration of MinD at poles, additional negative controls will add to the experiment.

1) Additional controls like any other membrane bound protein OR a MinD mutant defective in MinE binding will be a good addition.

2) Since oscillations are set by MinD's ATP hydrolysis, a non-hydrolysable ATP analog addition will also serve as a good control.

3) To test the importance MinE's membrane association, the authors could use a MinE deleted of its MTS.

It's shown that cardiolipin is not necessary and negative charge is sufficient to set MinDE oscillations. However, how important is negative charge in MinD patterning is not addressed.

The authors could try different lipid ratios to test if MinDE location and oscillations are affected.

The authors show that the MinCDE proteins are sufficient for positioning FtsZ to the medial region and excluding it from the pole ends. Again a few additional experiments to test the importance of MinE's membrane association & interaction with MinD in regulating FtsZ positioning could be added.

1) They could use a MinE MTS deletion and MinD interaction mutants.

2) It would also be interesting to see the effect of at least 2 mutations affecting membrane binding that Lutkenhaus lab had identified (Cell, Volume 146, Issue 3, 5 August 2011, Pages 396-407) in MinE on the oscillations and FtsZ positioning.

3) Although it shown that cardiolipin is not necessary for the oscillations, it is possible that cardiolipin has direct effects on FtsZ filaments. It may therefore play a role in FtsZ ring organization and positioning. To test if cardiolipin plays a role in medial localization of FtsZ, one could test FtsZ position in the presence of other lipids as done in earlier experiments.

Does the period of pole-to-pole oscillation change with increasing lengths? This is not indicated in the paper.

An interesting experiment that the authors do is to mimic the septating cell and find double oscillations similar to the *in vivo* conditions. An extreme extension of the same could be to mimic an ftsZ84ts strain after shift up shift down, where multiple septa are formed. Would oscillations happen between each constricting septa.

In the discussion, the authors speculate that gradients across the short axis might determine the division axis in a bacterium like Laxus oneistus. This is a very interesting idea given the presence of Min system in this bacterium. However, the authors should a statement speculating what factors could lead to gradients in short axis even in widths as small as 0.5 um as against the large widths (>10um) in the experiments performed.

*Reviewer 2*:

The fascinating oscillation of MinD/E in *E. coli* was discovered 15 years ago and has been of major interest to experimentalists, physicists and mathematicians ever since. In 2008 the Schwille lab reported a major advance by reconstituting the MinD/E system on planar lipid bilayers, where the two proteins interacted to generate moving waves. Their proposed reaction-diffusion model has provided insight into the mechanism, and they have explored the system in several subsequent papers. In 2013 Zieske and Schwille described creation of cylindrical PDMS chambers coated with lipid, in which the also reconstituted the MinD/E. Here the waves mimicked the oscillations obtained in *E. coli*.

The present paper used this PDMS chamber reconstitution in a much more thorough exploration. They determined that the charge of the lipid is the most important parameter, and they explored the co-localization of MinD and MinE (the previous study labeled and imaged only MinE). Most important they added FtsZ to the system, and showed that the oscillating Min compressed its localization to the middle of the chamber, similar to the mechanism *in vivo*. This is an important advance, which I think merits publication. I do have a couple of general issues that I think need to be addressed. Also, I found the paper to be lacking a surprising number of essential experimental details, which marred the presentation and made understanding difficult. In addition to the specific issues I raise, the authors should make a much stronger effort for a complete and polished presentation.

1) One general issue is that the chambers seem to operate with the same aspect ratio as they do in *E. coli,* yet they are on a ten times larger scale. I think this begs some detailed comparison with the system in *E. coli*, especially in terms of speed of movement of the waves, and oscillation period. The graphs in 1s1e, showing co-localization of MinD and E, are potentially interesting. Can these be compared with *E. coli*? Can it be related to the localization on planar lipid bilayers?

2) Another issue is the limited restriction of FtsZ in the present system. In *E. coli* the Min system seems to limit FtsZ to a single sharp Z ring localized precisely in the mid cell. In the present system it is a fuzzy patch 5+ µm wide. Is this because of the 10X larger scale? This deficiency of the present system at least needs to be noted.

3) A final general concern is that the previous Arum gam study concluded that curvatures ∼1 µm diameter caused optimal perpendicular alignment of pfs, while the alignment was lost for diameters ∼3-5 µm. Yet in the 10 µm chambers here they appear nicely aligned. Also, from the Schweizer study one can conclude that the direction of oscillation was established by width and length of flat lipid patches, so the curvature of the present chambers is probably not playing a role in the present oscillations. These two points should be addressed briefly.

Technical and editorial concerns:

4) I was confused about the construction of the chambers. From the EM they appear to be hemi cylinders, although the circular profile of the cylinder might not be exact. My confusion is what goes on top? Is there a cover slip? In this case the profile is a half circle with a flat top. The Methods have a somewhat obscure statement “the cured bulk PDMS was carefully peeled off and then the glass cover slide with an about 30 μm high microstructured PDMS layer were peeled off with the help of a razor blade.” (The two peeled are confusing.) That sounds like the PDMS is separated, but I would imagine it has to be sealed again with a cover slip. Is the cover slip separately coated with lipid? And then finally sealed? In the caption to Figure 2 the statement means nothing to me: “The buffer was not yet reduced to a level below the upper level of the chip. Therefore membrane and FtsZ-mts network are still intact on the upper level of the chip.” The Zieske ref provides a clue. It suggests that the PDMS chambers are simply open on top, no cover slip. A liquid-air interface. This needs more detailed description, including when and how the liquid was lowered (this for each figure). If the setup is like Zieske, I think a diagram like that 1b should be added, and perhaps also a cross section of the chamber (indicating its hemicircular shape, if that is reasonably established.

It is stated that a 25X/0.8 lens was used for some images – this would be suitable for the air-water interface. A 40X/1.20 objective was used for some images. This would be water immersion, very different from the liquid air interface. These experimental details should be given for each image.

5) In Figure 4 there are various constrictions in the center of the trough. We need some indication of the shape of the constriction relative to the hemi-cylinder and presumably flat surface of the liquid-air interface on top. Can any estimate be given for the diameter at the constriction?

“phosphatidylethanolamine (PE), phosphatidylglycerol (PG) and cardiolipin (CA), which have a net negative charge.” I think only the last two are negatively charged.

6) I found an apparent contradiction in the sizes and scale bars. The width of the compartments is claimed in Results to be ∼10 µm. From the scale bars in Figure s1 I'd make them to be 2 or 5 µM (1a and 1b). Also in Figure 1 I'd make them 2 and 5 µm. In Video 1 the scale bar says 50 pixels. This needs to be given in microns, and made to agree with all the above. All of these problems may be related to the confusion about the chamber shape, see above, and the level of focus. This needs to be clarified and stated for all cases.

7) More details are needed for how the time-averaged fluorescence intensity is calculated, or more specifically the value for a given time. One possibility would be to take a single image focused on the middle of the compartment (but see my questions above about the shape of the compartment - would that be the top surface at the air water interface, or lower?). Alternatively one would do a Z stack and determine the total concentration at each point on the long axis, and somehow average over the hemi-circle.

8) In Figure 2, what Min is labeled? In Figure 3 what Min is labeled? I suppose an intelligent reader can deduce that FtsZ is blue, but this should be taken care of by the authors. Is the FtsZ-mts the same or similar to the construct of Osawa and Erickson? If so a reference is needed. If not details of the construct should be given, e.g., what is the sequence of mts?

*Reviewer 3*:

This is another paper in an ongoing series from the Schwille lab to reproduce dynamic pattern formation in an *in vitro* system that mimics *in vivo* behavior. This paper flows from results presented in 2 prior papers. One describes the oscillation of MinD and MinE in a cylindrical compartment (50). Another is a paper that shows that FtsZ filaments are dynamically positioned on a flat surface where Min proteins are at a minimum (2). In this follow up paper the authors exploit the system developed in the 2013 paper to show that the Min proteins can position FtsZ within micro compartments that have the shape of a cell. This positioning of FtsZ in these micro compartments is another step in the bottom up approach to reconstitute bacterial cell division. It is a nice demonstration although it is predictable based upon there prior two publications. It would be improved if the authors characterized the MinC contribution to FtsZ localization in more detail to confirm the effect of MinC.

---

## [Author Response]

Throughout the manuscript, we now provided more experimental details and controls. In particular we now provide figures of the compartment profiles (Figure 1—figure supplement 1) and added a new figure (Figure 1—figure supplement 2), which visualizes the fabrication process of the samples.

*As you will see the referees have raised a number of points, a large number of which pertain to providing more details of experimental details and to some additional controls. However, one major concern that we would like you to further address is the role of MinC in FtsZ ring positioning in your studies. For example, is it possible to assemble more focused rings if full length MinC were used? The experiments should also be performed with the MinC-G10D mutant to correlate the results of the in vitro experiments with in vivo experiments. We would like you to consider and respond to the other comments made by the referees, if possible with experiments or with explanations to rebut the points raised. With the appropriate changes, your work will make a very important contribution to the fields of bacterial cell biology and cytokinesis in general*.

We now provide more experimental details and additional controls, as suggested by the reviewers. The employed MinC construct harbors all MinC domains. We used the plasmid described previously (Loose M, *et al.*, *Nat Struct Mol Biol.* 18, 577-83 (2011)) for overexpression of MinC. This plasmid codes for an open reading frame for MinC connected to an N-terminal hexahistidine tag by a linker.

Reviewer 1:

*[…] MinDE oscillations are shown in*
Figure 1*. While it’s clear that MinD and MinE are sufficient to set the oscillations and achieve the maximum concentration of MinD at poles, additional negative controls will add to the experiment*.

*1) Additional controls like any other membrane bound protein OR a MinD mutant defective in MinE binding will be a good addition*.

We have now performed a set of additional negative controls, which are shown in a new figure (Figure 2—figure supplement 1). Reconstituting MinE with another membrane bound protein (FtsZ-mts, which attached to membranes by the MinD membrane targeting sequence) confirmed that MinE was not recruited to the membrane when reconstituted with a different protein than MinD (Figure 2—figure supplement 1).

*2) Since oscillations are set by MinD's ATP hydrolysis, a non-hydrolysable ATP analog addition will also serve as a good control*.

The suggested experiment has been performed by our group previously. It has been shown that purified and fluorescently labeled MinD and MinE do not form patterns on membranes in vitro if the nonhydrolysable ATP analog ATPγS was used instead of ATP (Loose M, *et al.*, *Science* 320, 789-792 (2008)). We now mention this experiment in the main text: “Previous controls with purified Min proteins on flat bilayers and the non-hydrolysable ATP analog ATPγS confirmed that the dynamic protein patterns were thereby powered by ATP (Loose M, *et al.*, *Science* 320, 789-792 (2008)).”

*3) To test the importance MinE's membrane association, the authors could use a MinE deleted of its MTS*.

We now demonstrate the importance of MinE’s membrane association by using MinE deleted of its membrane targeting sequence. In particular, we demonstrated that the Min system is unable to form a stable gradient with the concentration minimum in the middle of the cell without MinE’s membrane association (Figure 1—figure supplement 5).

It's shown that cardiolipin is not necessary and negative charge is sufficient to set MinDE oscillations. However, how important is negative charge in MinD patterning is not addressed.

*The authors could try different lipid ratios to test if MinDE location and oscillations are affected*.

In Figure 1—figure supplement 4 we demonstrate that different lipid ratios affect the spatial scales of Min protein patterns. Since the microcompartments were adjusted to the spatial scales of Min pattern on *E. coli* membranes, no stable oscillations would be formed in the compartments. We now discussed this point and the importance of the right amount of negative charge in the main text. “For high amounts of negatively charged lipids, the length scales of the protein patterns decrease (Figure 1—figure supplement 4 and (Vecchiarelli AG, *et al.* Mol Microbiol. 93, 453-63 (2014)). Thus, the spatial scales of the Min proteins on highly negatively charged membranes would be too small to form stable pole-to-pole oscillations in compartments, which were engineered for protein length scales on *E. coli* membranes. The implication that Min protein oscillations are perturbed if the membrane charge increases is in agreement with live cell studies of GFP-MinD in PE-lacking *E. coli* cells, which demonstrate that Min protein pattern are affected if the membrane charge is (29) and provides further evidence that a balanced ratio of charged and non-charged lipids is of critical importance for the live cycle of *E. coli*.”

*The authors show that the MinCDE proteins are sufficient for positioning FtsZ to the medial region and excluding it from the pole ends. Again a few additional experiments to test the importance of MinE's membrane association & interaction with MinD in regulating FtsZ positioning could be added*.

*1) They could use a MinE MTS deletion and MinD interaction mutants*.

*2) It would also be interesting to see the effect of at least 2 mutations affecting membrane binding that Lutkenhaus lab had identified (Cell, Volume 146, Issue 3, 5 August 2011, Pages 396-407) in MinE on the oscillations and FtsZ positioning*.

We agree that the role of MinE’s membrane association and its interaction with MinD are highly interesting areas which should be addressed with reconstitution techniques in order to complement existing experiments of the Lutkenhaus lab (Cell, Volume 146, Issue 3, 5 August 2011, Pages 396-407). Using a MinE MTS deletion, we now demonstrate that bottom-up experiments are indeed able to provide new insights into the importance of the MTS (Figure 1—figure supplement 5).

However, a study involving a thorough biochemical characterization, reconstitution and analysis of mutants in a synthetic system to address the role of individual aminoacids and functional domains of these proteins represents a whole project with data for a full paper on its own. We are working on such a project, but the finalization of this project requires much more time. We thank the reviewer for addressing the potential of our bottom-up system to study Min mutants, and we are positive that questions regarding the importance of individual functional domains of MinE will be addressed in a future manuscript.

*3) Although it shown that cardiolipin is not necessary for the oscillations, it is possible that cardiolipin has direct effects on FtsZ filaments. It may therefore play a role in FtsZ ring organization and positioning. To test if cardiolipin plays a role in medial localization of FtsZ, one could test FtsZ position in the presence of other lipids as done in earlier experiments*.

In *Escherichia coli* FtsZ is attached to the membrane by FtsA and ZipA. Here, we used an FtsZ fusion with a membrane targeting sequence (mts). While the mts fulfills the role to attach FtsZ filaments to the membrane, it is unclear how the characterization of the membrane interaction of the mts in dependence of the lipid composition can be compared to the natural membrane anchors (FtsA and ZipA) of FtsZ. The possibility to gain insights into a physiologically relevant role of lipid composition for medial FtsZ is therefore limited with our bottom-up system.

To address the question of the reviewer, reconstitution assays involving Min proteins and the natural membrane anchoring proteins of FtsZ need to be developed. This work is in progress in the lab.

*Does the period of pole-to-pole oscillation change with increasing lengths? This is not indicated in the paper*.

We now included data how the oscillation period changes with increasing length (Figure 4) to address this question, described in the section starting “While the length dependent increase of oscillation modes was well conserved in our experiments, the oscillation period was prone to small perturbation…”

*An interesting experiment that the authors do is to mimic the septating cell and find double oscillations similar to the in vivo conditions. An extreme extension of the same could be to mimic an ftsZ84ts strain after shift up shift down, where multiple septa are formed. Would oscillations happen between each constricting septa*.

We addressed the reviewers question experimentally and designed compartments with two septa. Figure 6—figure supplement 3 now demonstrates that oscillations happen between each “constricting septum”.

*In the discussion, the authors speculate that gradients across the short axis might determine the division axis in a bacterium like Laxus oneistus. This is a very interesting idea given the presence of Min system in this bacterium. However, the authors should a statement speculating what factors could lead to gradients in short axis even in widths as small as 0.5 um as against the large widths (>10um) in the experiments performed*.

The Min patterns *in vivo* are about ten times smaller than *in vitro*. We now included more information on parameters could account for the difference in the scales in living bacteria vs. *in vitro* systems. “Compared to living cells, the temporal scale for the oscillation period is conserved in the synthetic system. However, both the compartments and the spatial scales of the protein waves are about ten times bigger as compared to living cells. Why the Min patterns *in vivo* are about ten times larger than *in vitro* is quantitatively not fully understood. Different protein ratios, ionic strength of the buffer and membrane composition have been shown to affect the spatial scales (25, 48). Furthermore, the membrane potential (45) and molecular crowding in a living cell could influence Min protein patterns.”

Furthermore, we speculate that on a bacterial scale gradients along the short axis might be regulated by parameters such as cell width, reaction- and diffusion rates:

“It is also unknown whether a similar gradient along the short axis of the cell is used in living cells, and which factors might contribute on the bacterial scale to regulate gradients along the short axis. In line with our *in vitro* results, Min oscillations along the short axis of a bacterial cell might be accomplished by an increased width as compared to an *E. coli* cell, or by smaller scales of Min protein patterns through modified reaction and/or diffusion rates of the proteins. However, also other systems which generate gradients along the length axis of bacteria cell, such as activity gradients of phosphorylated proteins (5), might provide mechanisms to generate gradients along the short axis of a bacterial cell.”

Reviewer 2:

*[…] The present paper used this PDMS chamber reconstitution in a much more thorough exploration. They determined that the charge of the lipid is the most important parameter, and they explored the co-localization of MinD and MinE (the previous study labeled and imaged only MinE). Most important they added FtsZ to the system, and showed that the oscillating Min compressed its localization to the middle of the chamber, similar to the mechanism in vivo. This is an important advance, which I think merits publication. I do have a couple of general issues that I think need to be addressed. Also, I found the paper to be lacking a surprising number of essential experimental details, which marred the presentation and made understanding difficult. In addition to the specific issues I raise, the authors should make a much stronger effort for a complete and polished presentation*.

The revised manuscript now contains more experimental details, a new figure that depicts the process of sample preparation (Figure 1—figure supplement 2) and images to give more insight into the geometry of the micro compartments (Figure 1—figure supplement 1).

*1) One general issue is that the chambers seem to operate with the same aspect ratio as they do in* E. coli*, yet they are on a ten times larger scale. I think this begs some detailed comparison with the system in* E. coli*, especially in terms of speed of movement of the waves, and oscillation period*.

The spatial scale of the patterns is about ten times larger *in vitro*. The temporal scale of the oscillations is comparable *in vivo* and *in vitro*. Thus, the oscillation period is comparable *in vivo* and *in vitro* and the speed of the pattern is about ten times larger. We now included comparisons of the temporal and spatial scales in the synthetic system vs. *E. coli*:

Comparison of spatial scales with *E. coli*: “Compared to living cells, the temporal scale for the oscillation period is conserved in the synthetic system. However, both the compartments and the spatial scales of the protein waves are about ten times bigger as compared to living cells. Why the Min patterns *in vivo* are about ten times larger than *in vitro* is quantitatively not fully understood. Different protein ratios, ionic strength of the buffer and membrane composition have been shown to affect the spatial scales (25, 48). Furthermore the membrane potential (45) and molecular crowding in a living cell could influence Min protein patterns.”

Comparison of temporal scales with *E. coli*: “oscillation periods between less than a minute and 4 minutes were measured. This time scale is well in agreement with measurements of 0.5 to 2 minutes for the oscillation period in living cells (39).”

*The graphs in 1s1e, showing co-localization of MinD and E, are potentially interesting*. *Can these be compared with E. coli? Can it be related to the localization on planar lipid bilayers?*

We now compared the results with *E. coli*: “Although the size of bacterial cells and photo bleaching is limiting for characterizing the Min profiles *in vivo* in such detail as it is possible *in vitro*, the intensity peak of MinE at the trail of the Min pattern was also observed in *E. coli* and is generally referred to as ‘MinE ring’.”

Part of the dynamics in microcompartments can be related to Min dynamics on planar bilayers: “This disassembly of Min protein patterns at the trailing zone during each oscillation is comparable to the disassembly at the trailing edge of travelling Min patterns on flat membranes. The main difference of Min proteins profiles on flat membranes and in microcompartments is that the concentration profiles on flat membranes only change their localization while traveling across the membrane. In contrast, during oscillations in compartments new membrane attached Min protein patterns need to be repeatedly established and disassembled every half oscillation period, which results in a remarkable remodelling of the concentration profiles during the oscillation cycle and an asymmetry in their rise and decay phases at the poles.”

*2) Another issue is the limited restriction of FtsZ in the present system. In* E. coli *the Min system seems to limit FtsZ to a single sharp Z ring localized precisely in the mid cell. In the present system it is a fuzzy patch 5+ µm wide. Is this because of the 10X larger scale? This deficiency of the present system at least needs to be noted*.

We added the following paragraph: “In contrast to the living cell, the FtsZ-mts distribution in the middle of synthetic compartments appears to be rather broad. The major reason for the wide FtsZ-mts distribution is most likely the ten times larger Min protein pattern *in vitro,* as compared to living cells. The wide MinC minimum allows multiple FtsZ-mts bundles to localize in a wide region in the middle of the compartment.”

*3) A final general concern is that the previous Arum gam study concluded that curvatures ∼1 µm diameter caused optimal perpendicular alignment of pfs, while the alignment was lost for diameters ∼3-5 µm. Yet in the 10 µm chambers here they appear nicely aligned. Also, from the Schweizer study one can conclude that the direction of oscillation was established by width and length of flat lipid patches, so the curvature of the present chambers is probably not playing a role in the present oscillations. These two points should be addressed briefly*.

For the Min protein oscillations, the curvature indeed does not play a role. However, note that the geometrical requirements for aligning oscillating Min patterns in compartments and travelling Min pattern on flat membranes are different. The previous Schweizer et al. study did not demonstrate oscillating Min pattern, but investigated the alignment of traveling Min waves instead. As demonstrated in Figure 6—figure supplement 1 oscillatory Min patterns are not aligned in compartments with a width of about 40 μm while traveling waves are still aligned at a membrane widths as big as 100 μm.

We are sorry for the previously insufficient description of the chambers. We now included profiles of the chambers, which show that the chambers are 10 μm on the top and are tapered towards the bottom (Figure 1—figure supplement 1). We now also demonstrate that FtsZ-mts is aligned in tapered compartments as compared to compartments with a flat bottom (Figure 1—figure supplement 1).

*Technical and editorial concerns*:

*4) I was confused about the construction of the chambers. From the EM they appear to be hemi cylinders, although the circular profile of the cylinder might not be exact. My confusion is what goes on top? Is there a cover slip? In this case the profile is a half circle with a flat top*.

We now revised the presentation of the construction of the chambers and added much more information. To better describe the profile of the chambers we now supplemented Figure 1—figure supplement 1 with a profile image of the chambers (Figure 1—figure supplement 1). To provide a more detailed description of the construction of the chambers, the fabrication process is now depicted in a new figure (Figure 1—figure supplement 2) and explained in the corresponding figure caption and in the Materials and methods section. We now also point out that the microstructured PDMS compartments are open on the top and clad with membrane at the bottom and the sides. “The top of the chambers was open with a buffer/air interface.”

*The Methods have a somewhat obscure statement “the cured bulk PDMS was carefully peeled off and then the glass cover slide with an about 30 μm high microstructured PDMS layer were peeled off with the help of a razor blade.” (The two peeled are confusing.) That sounds like the PDMS is separated, but I would imagine it has to be sealed again with a cover slip. Is the cover slip separately coated with lipid? And then finally sealed*?

There are indeed two layers of PDMS involved in the fabrication process (a thick bulk PDMS layer and a thin PDMS layer harboring the microstructures), but we see that the description might have been confusing.

To clarify the fabrication process of the PDMS microstructures, we now included a photograph and a schematic image of the Wafer/PDMS/glass cover slip/PDMS sandwich (Figure 1—figure supplement 2).

In the Methods section, we changed the description and included a reference that points to the figure describing the fabrication process in more detail. “After curing the PDMS at 80°C overnight, the bulk PDMS layer was peeled off. With the help of a razor blade, the glass coverslide with the about 30μm thin, micro structured PDMS layer was carefully separated from the wafer and used as sample support (Figure 1—figure supplement 2).”

*In the caption to*
Figure 2
*the statement means nothing to me: “The buffer was not yet reduced to a level below the upper level of the chip. Therefore membrane and FtsZ-mts network are still intact on the upper level of the chip.” The Zieske ref provides a clue. It suggests that the PDMS chambers are simply open on top, no cover slip. A liquid-air interface. This needs more detailed description, including when and how the liquid was lowered (this for each figure). If the setup is like Zieske, I think a diagram like that 1b should be added, and perhaps also a cross section of the chamber (indicating its hemicircular shape, if that is reasonably established*.

The reduction of the buffer to enclose proteins in compartments is now described in detail in Figure 1—figure supplement 2. This figure also contains a schematic diagram and a description at which step during the sample preparation and how the buffer reservoir was reduced. “Subsequently the buffer reservoir was manually reduced using a pipette, such that the buffer remained only in the micro fabricated chambers.”

The procedure when and how the liquid was lowered was the same for each figure with micro chambers, except from the figures where it is indicated that “The buffer was not yet reduced to a level below the upper level of the chip.” Cross sections of the chambers are now added in Figure 1—figure supplement 1.

*It is stated that a 25X/0.8 lens was used for some images – this would be suitable for the air-water interface. A 40X/1.20 objective was used for some images. This would be water immersion, very different from the liquid air interface. These experimental details should be given for each image*.

The microscope we used was an inverted confocal microscope. Thus, the objective was not positioned on top of the sample, but the objective in our setup was positioned below the sample and the protein patterns were imaged through the transparent glass/PDMS layer. For such an arrangement of the setup both objectives are appropriate.

The use of one objective or the other was dependent on the availability of two LSM780 setups at the institute which were equipped with different objectives. In particular, either of the two objectives could be used for all experiments performed for this study.

We now included the information that the microscope is inverted to prevent confusions: “Image acquisition was performed on a ZEISS (Jena, Germany) LSM780 confocal laser scanning microscope (inverted microscope)…”

*5) In*
Figure 4
*there are various constrictions in the center of the trough. We need some indication of the shape of the constriction relative to the hemi-cylinder and presumably flat surface of the liquid-air interface on top. Can any estimate be given for the diameter at the constriction*?

We now included a picture of the septum profile in Figure 6—figure supplement 3. Moreover, the microstructure is now described in more detail in the Materials and methods section “Design of microcompartments”.

*“phosphatidylethanolamine (PE), phosphatidylglycerol (PG) and cardiolipin (CA), which have a net negative charge.” I think only the last two are negatively charged*.

This sentence was meant to point out that a mixture of these three lipids is negatively charged in the sum. To prevent misunderstandings, we changed the sentence to “Note that the *E. coli* lipid extract for generating membranes in these experiments is a mixture of mainly three different lipids, (phosphatidylethanolamine (PE, neutral), phosphatidylglycerol (PG, negatively charged) and cardiolipin (CA, negatively charged)) and that the membranes have a net negative charge.”

*6) I found an apparent contradiction in the sizes and scale bars. The width of the compartments is claimed in Results to be ∼10 µm. From the scale bars in Figure s1 I'd make them to be 2 or 5 µM (1a and 1b). Also in*
Figure 1
*I'd make them 2 and 5 µm*.

The scale bar for the electron micrographs is only valid for the x-axis of the picture. We are now mentioning this in the figure caption. “Note that the scale bars in the electron micrographs only applies to the x-direction of the picture. The sample was tilted in the other direction to create the three-dimensional impression of electron micrographs.”

The micro structures are tapered towards the bottom of the structures. The “10μm” refer to the upper rim of the structures. Thus, in images with the focal plane at the bottom of the compartment, the borders appear narrower than 10 μm. To prevent confusions about the scales of the microstructures we now included profiles in Figure 1—figure supplement 1.

*In*
Video 1
*the scale bar says 50 pixels. This needs to be given in microns, and made to agree with all the above. All of these problems may be related to the confusion about the chamber shape, see above, and the level of focus. This needs to be clarified and stated for all cases*.

Corrected. More details about the chamber shape are now given throughout the manuscript.

*7) More details are needed for how the time-averaged fluorescence intensity is calculated, or more specifically the value for a given time. One possibility would be to take a single image focused on the middle of the compartment (but see my questions above about the shape of the compartment - would that be the top surface at the air water interface, or lower?). Alternatively one would do a Z stack and determine the total concentration at each point on the long axis, and somehow average over the hemi-circle*.

We now included the following sentence: “The time averaged distribution of protein concentrations was calculated by acquiring time-lapse-images with the focal plane at the middle of the compartment and averaging the intensity of the acquired frames.”

*8) In*
Figure 2*, what Min is labeled*?

The labeled protein is now mentioned in the figure caption: “When the gradient forming Min system (red: MinE.Atto655) and FtsZ (blue) are reconstituted together, the Min gradient coordinates the localization of FtsZ-mts to the middle of the compartments.”

*In*
Figure 3
*what Min is labeled? I suppose an intelligent reader can deduce that FtsZ is blue, but this should be taken care of by the authors*.

The corresponding proteins for the red and blue color are now indicated in the figure caption: “In compartments with double oscillations of the Min system (red: MinE.Atto655), FtsZ-mts (blue) localizes to two regions along the cell length.”

*Is the FtsZ-mts the same or similar to the construct of Osawa and Erickson? If so a reference is needed. If not details of the construct should be given, e.g., what is the sequence of mts*?

The FtsZ-mts construct is the same construct of Osawa and Erickson. We apologize for the lack of the original reference. We now added the reference in the main text (“To keep the system as minimal as possible, we avoided the native membrane adaptors of FtsZ, FtsA and ZipA, and used a FtsZ hybrid protein, fused to YFP and a membrane targeting sequence (FtsZ-mts) (31)”) as well as in the Materials and methods section (“Purification of FtsZ-YFP-mts was originally described by the Erickson group (31))”.

Reviewer 3:

*[…] In this follow up paper the authors exploit the system developed in the 2013 paper to show that the Min proteins can position FtsZ within micro compartments that have the shape of a cell. This positioning of FtsZ in these micro compartments is another step in the bottom up approach to reconstitute bacterial cell division. It is a nice demonstration although it is predictable based upon there prior two publications. It would be improved if the authors characterized the MinC contribution to FtsZ localization in more detail to confirm the effect of MinC*.

We are grateful for the reviewers’ suggestions to improve the characterization of how exactly MinC positions FtsZ, in particular, by testing the MinC-G10D mutant in vitro (see below). We believe that these additional experiments greatly improved the manuscript. The MinC-G10D mutant (Figure 3) was unable to position FtsZ-mts, indicating that mainly the N-terminal domain of MinC is involved in inhibiting FtsZ in our *in vitro* system. Furthermore, we now included additional controls to confirm the effect of MinC (Figure 2—figure supplement 1).